# Topological isolation of developmental regulators in mammalian genomes

Hua-Jun Wu[1,2,3,12], Alexandro Landshammer [4,5,12], Elena K. Stamenova[6], Adriano Bolondi[4,5], Helene Kretzmer[4], Alexander Meissner [4,5,6,7✉] & Franziska Michor [6,7,8,9,10,11✉]

Precise control of mammalian gene expression is facilitated through epigenetic mechanisms and nuclear organization. In particular, insulated chromosome structures are important for regulatory control, but the phenotypic consequences of their boundary disruption on developmental processes are complex and remain insufficiently understood. Here, we generated deeply sequenced Hi-C data for human pluripotent stem cells (hPSCs) that allowed us to identify CTCF loop domains that have highly conserved boundary CTCF sites and show a notable enrichment of individual developmental regulators. Importantly, perturbation of such a boundary in hPSCs interfered with proper differentiation through deregulated distal enhancer-promoter activity. Finally, we found that germline variations affecting such boundaries are subject to purifying selection and are underrepresented in the human population. Taken together, our findings highlight the importance of developmental gene isolation through chromosomal folding structures as a mechanism to ensure their proper expression.

[1] Center for Precision Medicine Multi-Omics Research, Peking University Health Science Center, Beijing, China. [2] School of Basic Medical Sciences, Peking University Health Science Center, Beijing, China. [3] Peking University Cancer Hospital and Institute, Beijing, China. [4] Department of Genome Regulation, Max Planck Institute for Molecular Genetics, Berlin, Germany. [5] Institute of Chemistry and Biochemistry, Freie Universität Berlin, Berlin, Germany. [6] The Broad Institute of MIT and Harvard, Cambridge, MA, USA. [7] Department of Stem Cell and Regenerative Biology, Harvard University, Cambridge, MA, USA. [8] Department of Data Science, Dana-Farber Cancer Institute, Boston, MA, USA. [9] Department of Biostatistics, Harvard T. H. Chan School of Public Health, Boston, MA, USA. [10] The Ludwig Center at Harvard, Boston, MA, USA. [11] Center for Cancer Evolution, Dana-Farber Cancer Institute, Boston, MA, USA. [12]These authors contributed equally: Hua-Jun Wu, Alexandro Landshammer. ✉email: meissner@molgen.mpg.de; michor@jimmy.harvard.edu

Mammalian genomes are organized into a hierarchy of local structures including megabase-sized topologically associating domains (TADs) and DNA loops that are usually localized within TADs[1–10]. The majority of TADs are stable across cell types and conserved between mouse and human[2,8], while DNA loops of enhancer-gene interactions are generally more celltype-specific[11,12]. Recent studies identified specific DNA loops organized by CCCTC-binding factor (CTCF) and cohesin; these loops, called insulated neighborhoods, are local structures within TADs that encompass most enhancer-promoter loops[5].

Disrupting the boundaries of TADs or insulated neighborhoods can lead to novel chromosomal interactions and ectopic long-range enhancer adoption, which can interrupt key gene function[3,13]. Such altered boundary elements, usually caused by structural variations, can also lead to developmental disorders in humans. The first human to mouse translational case study focused on abnormal limb syndromes caused by genomic alterations at the TAD boundaries containing the *EPHA4* locus. In this study, a cluster of limb enhancers normally associated with the *Epha4* gene was found to be misplaced and to ectopically activate genes, including *Wnt6*, *Pax3*, and *Ihh*, in the neighboring TADs[14]. A related study showed that genomic duplication of a murine boundary between *Kcnj2* and *Sox9* TADs resulted in the formation of new TADs, the ectopic activation of *Kcnj2* and the onset of Cooks Syndrome—another limb malformation[15]. However, duplication of smaller DNA segments at the same locus within the *SOX9* TAD causes a different phenotype, that of sex reversal, in humans[16]. Moreover, different chromosomal conformations of the *Pitx1* locus have been shown to lead to activation of *Pitx1* by ectopic interactions with its active enhancer Pen in the forelimb, causing partial arm-to-leg transformation in both human and mouse[17]. In another example, a large genomic deletion leading to enhancer adoption by the *LMNB1* gene was identified as an alternative path to autosomal dominant adult-onset demyelinating leukodystrophy[18]. In addition, in 273 subjects with congenital anomalies, 7.3% of balanced chromosomal abnormalities (BCAs) disrupted TADs containing known syndromic loci; for instance, breakpoints of BCAs in eight subjects disrupted the *MEF2C*-containing TAD, resulting in decreased expression of *MEF2C*, which is linked to 5q14.3 microdeletion syndrome[19].

These selected case studies suggest a crucial role for insulated chromosome structures in gene regulation, raising the question whether this is a more universal mechanism that contributes to precise gene control by limiting domain-level access to regulatory elements in development. One previous study demonstrated gene expression changes upon boundary disruptions in mouse ES cells[5], but it remains incompletely understood how insulation boundaries influence early stem and progenitor differentiation. Prior work has also demonstrated higher sequence conservation across primates[20] and elevated somatic mutation rates across tumor types[10] in the boundary CTCF motifs of insulation domains as compared to other sequences; however, it remains to be shown whether and how insulating structures shape the gene distribution across the human genome and whether there are associations between the function of an insulating domain and the number of genes it contains.

We hypothesized that key developmental regulators, especially factors directing early differentiation, might be shielded through insulated structures from nearby genes to facilitate local regulation, and that disruption of their boundaries might lead to deregulation and consequently cellular defects. To investigate this, we conducted a systematic study, analyzing patterns of insulated domains across the human genome and show that CTCF loop domains[21,22] display an intriguing enrichment that may facilitate proper regulation of crucial genes. Specifically, we find that early developmental regulators appear preferentially isolated from other genes, with looped boundary CTCF sites that are highly conserved across cell types. To functionally explore their role, we used single guide (sg)RNA Cas9-directed genome perturbation to disrupt the CTCF loop domain boundary at the *SOX17* (an example of a single isolated gene) and *NANOG* (an example of a multi-gene domain) loci. Notably, the boundary of *SOX17*, but not *NANOG*, appears necessary for proper function: disruption led to *SOX17* misexpression and a failure to differentiate into endoderm. Moreover, we found a subset of CTCF loop domains with constitutive boundaries across many cell types, which are also more conserved in sequence across species. Our findings add further support to the contribution of gene insulation through chromosomal folding structures to enable defined expression of developmental regulators, especially during early differentiation.

## Results

**Identification of topologically insulated regions**. To explore the role of topological genome organization in pluripotent stem cells, we performed Hi-C experiments[1] in the human embryonic stem (ES) cell line HUES64 and generated a total of 1.05 billion uniquely mapped paired-end reads (Supplementary Data 1). These data led to the identification of 231,970 high-confidence interactions in the genomic range of 20 kb–2 Mb using Fit-Hi-C[23] (Methods). By mapping these interactions to CTCF consensus motifs, we obtained 37,428 significant CTCF-CTCF loops. Loops close to each other were clustered and merged to limit redundancy, yielding a total of 24,056 CTCF loop domains with a median length of 304 kb (Fig. 1a, Supplementary Fig. 1a and Supplementary Data 2). To further validate our domain calling results, we compared the identified loops to those reported by other Hi-C loop detection methods[24–26] and observed a large degree of agreement between Fit-Hi-C and HiCCUPS results (Supplementary Fig. 1b). Moreover, the CTCF loop domains identified by Fit-Hi-C depicted the highest agreement with the insulated neighborhoods identified by cohesin ChIA-PET data in primed human ES cells[20] (Supplementary Fig. 1c). We then calculated a directionality index, which provides a quantification for the degree of upstream or downstream interaction bias of a genomic region[2], for all CTCF loop domains and surrounding regions to demonstrate their topological insulation function (Supplementary Fig. 1d). Surprisingly, using this data, we found that many CTCF loop domains contain only a single protein-coding gene (~40%, $n = 9,673$) (Fig. 1b, 1c, and Supplementary Fig. 1e, 1f), which is a significant overrepresentation compared to what is expected (permutation test $p < 0.001$ when randomly shuffling either domains or genes, Fig. 1d, e). These CTCF loop domains are termed single-gene domains, and the genes contained are referred to as topologically isolated genes (TIGs). The remaining CTCF loop domains embed either multiple genes (38%, $n = 9,189$; Supplementary Fig. 1e) or no genes (22%, $n = 5,194$; Supplementary Fig. 1f), respectively.

**CTCF loop domain boundaries are largely preserved during ES cell differentiation**. We next sought to explore the stability of CTCF loop domain boundaries using an ES cell differentiation model and generated Hi-C data from ES-cell-derived endoderm (dEN), ectoderm (dEC), and mesendoderm (dMS). We found that these boundaries are well preserved during ES cell differentiation as exemplified at the *SOX17*, *SMAD1*, *SOX2*, and *NANOG* loci by visualizing the heatmaps and arc plots of Hi-C interactions (Supplementary Fig. 2a); however, this approach is not suitable for analyzing large numbers of boundaries at the

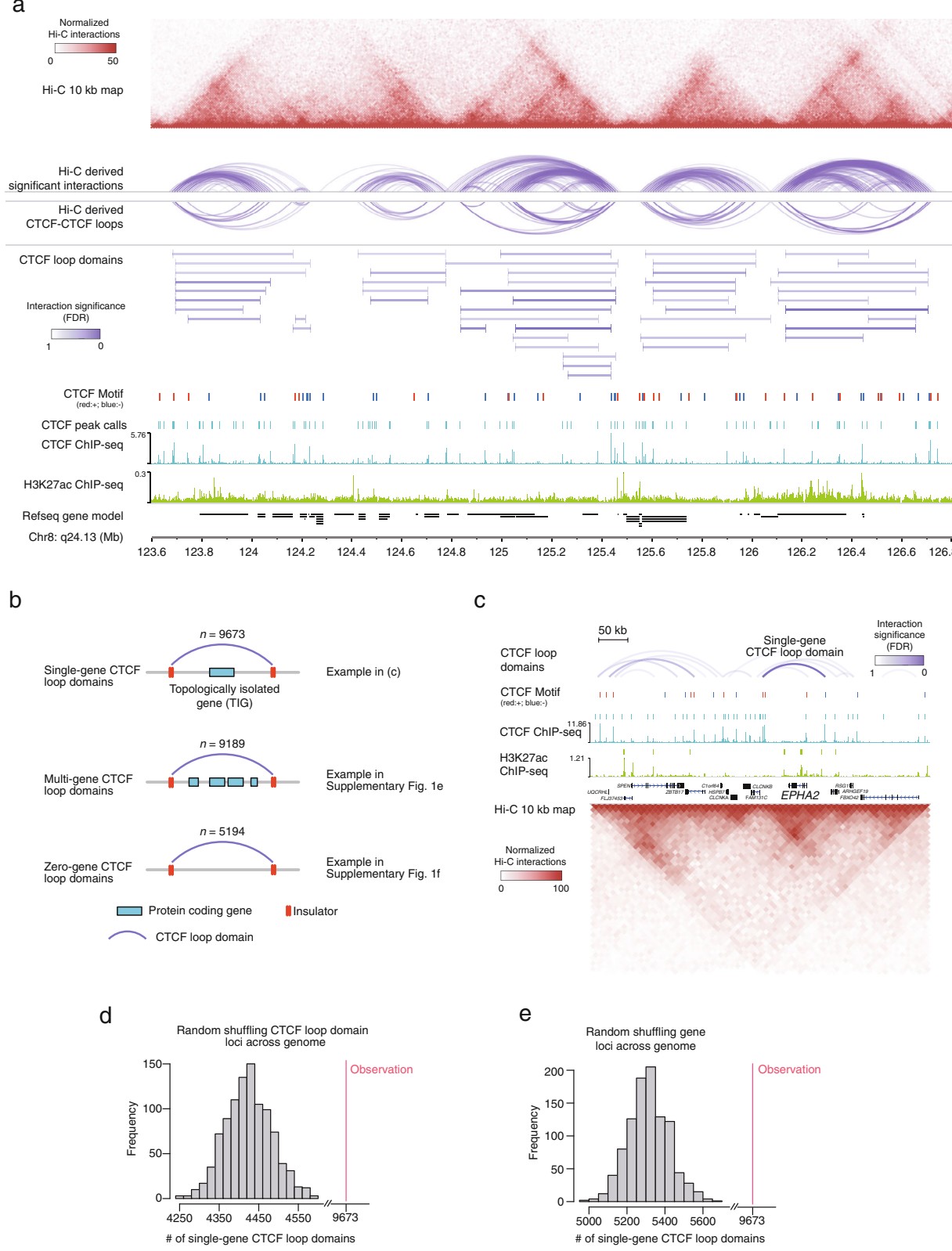

same time. We therefore used boundary-anchored virtual 4 C plots to visualize contact interactions between all pairs of boundaries in a sample. This approach uses two heatmaps, with the left heatmap representing the Hi-C interactions from the surrounding genomic regions of the left boundaries to right boundaries (similar to setting the left boundaries as viewpoints in

4 C data) and the right heatmap representing the Hi-C interactions from the surrounding genomic regions of the right boundaries to the left boundaries (Fig. 2a). For cases in which there are physical interactions between the two boundaries in the sample, the plot exhibits a high intensity in the center of both heatmaps but not the surrounding regions. By investigating the

**Fig. 1 Topologically insulated regions in HUES64 ES cells. a** Normalized Hi-C interaction map, high-confidence interactions (arc), CTCF-CTCF loops (arc), and CTCF loop domains (line) displayed on top of ChIP-seq profiles of CTCF and H3K27ac at chromosome 8q24.13 region. CTCF peaks are denoted by bars above the ChIP-seq profiles of CTCF. CTCF consensus motifs are denoted by red (forward orientation) and blue (reverse orientation) bars above the CTCF peaks. Normalized Hi-C interactions are shown as a heatmap with each pixel representing a 25 kb genomic region. The interaction significance (FDR) was calculated from Hi-C data. **b** Illustration of single-gene CTCF loop domains, multi-gene domains, and zero-gene domains. The numbers of domains in each group within HUES64 are displayed on top of each plot. **c** Display of a single-gene CTCF loop domain and the topologically isolated gene (TIG) at the *EPHA2* locus. CTCF consensus motifs, ChIP-seq profiles of CTCF and H3K27ac, and normalized Hi-C interaction maps are displayed. CTCF peaks and enhancers are denoted by bars above the ChIP-seq profiles of CTCF and H3K27ac. CTCF consensus motifs are denoted by red (forward orientation) and blue (reverse orientation) bars above the CTCF peaks. Normalized Hi-C interactions are shown as a heatmap with each pixel representing a 10 kb genomic region. CTCF loop domains are displayed on the top and the interaction significance (FDR) was calculated from Hi-C data. **d** The distribution of the number of single-gene domains in the human genome by randomly shuffling the domain loci across the genome. The red line indicates the observed number of single-gene domains in the genome. **e** The distribution of the number of single-gene domains in the human genome by randomly shuffling the gene loci across the genome. The red line indicates the observed number of single-gene domains in the genome.

profiles of both *P*-values (Fig. 2a) and normalized Hi-C contacts (Fig. 2b), we found that the boundaries of CTCF loop domains are largely preserved throughout the process of ES cell differentiation. We then aggregated these heatmaps for each sample by column to generate a profile plot representing the average and standard deviation of the signal across all boundaries. This analysis depicts a clear peak in the boundary centers of both single-gene and multi-gene domains (Fig. 2c and Supplementary Fig. 2b), and demonstrates that single-gene domain boundaries are more preserved than multi-gene domain boundaries (Fig. 2d).

To interrogate when these boundaries are formed during embryonic development, we analyzed recently published Hi-C data from early human embryos, including samples of 2-cell, 8-cell, morula, and blastocyst stages, as well as a 6-week time point[27]. We found that CTCF loop domain boundaries are gradually established starting from the 8-cell stage, at the same time or shortly after zygotic genome activation (ZGA)[28] during which CTCF gene expression is also strongly induced[27], and are stable after the blastocyst stage (Fig. 2e). At the blastocyst stage and 6-week time point, the single-gene domain boundaries demonstrated more pronounced Hi-C interactions than the multi-gene domain boundaries, as observed in ES cells and their derivatives, independent of genomic distances (Supplementary Fig. 2c). However, this difference was not observed in the early stages (8-cell and morula), implying that this boundary divergence arises between the morula and blastocyst stages and coincides with the initiation of lineage specification[29]. ZGA inhibition by α-amanitin (an RNA Pol II inhibitor) treatment was shown to also repress CTCF expression in the 8-cell stage[27]. Interestingly, α-amanitin treatment of 8-cell embryos had less influence on single-gene domain boundaries than on multi-gene domain boundaries (Fig. 2e). Once formed, it appears that single-gene domain boundaries are maintained independently of ZGA and of CTCF expression, and might thus be more stable and robust across diverse cellular processes.

**Developmental regulators are insulated by conserved CTCF boundaries**. The preservation of single-gene domain boundaries in ES cell differentiation may imply some functional importance. Indeed, further analysis demonstrated that TIGs are enriched for diverse developmental processes in the Gene Ontology (GO) database (Fig. 3a). Next, we defined developmental regulators as transcriptional factors under the GO term "developmental process" and performed enrichment analysis of the different CTCF loop domain groups, based on the number of genes they insulate. Interestingly, we found that developmental regulators are enriched for in single-gene CTCF loop domains, but less so in other domain groups with multiple genes (Fig. 3b). This association

motivated us to query whether the insulation function of these boundaries is important.

First, we analyzed the extent of conservation of these boundaries across different mammals and found that boundary CTCF motifs of single-gene CTCF loop domains are highly conserved across placental mammals (Fig. 3c and Supplementary Fig. 3a). This may suggest that these motifs are functionally important elements that undergo natural selection. In contrast, the boundary CTCF motifs of multi- and zero-gene CTCF loop domains have a sequentially decreasing conservation score, while CTCF motifs outside of any boundaries (nonboundary CTCF motifs) are generally not conserved (Fig. 3c and Supplementary Fig. 3a). We also observed that boundary CTCF sites of developmental regulator domains are more conserved than those of other genes (Fig. 3d). These analyses further support the notion that the boundary CTCF sites of single-gene domains, especially those insulating developmental regulators, may be functionally important. In addition, we found that boundaries of single-gene domains are more strongly interacting based on Hi-C data (Supplementary Fig. 3b) and are enriched for stronger CTCF-binding sites than other boundaries, with a similar average signal intensity for single- and multi-gene domain boundaries (Supplementary Fig. 3c-e). Finally, single-gene domains were found to contain more *cis*-regulatory elements per gene, such as enhancers and long noncoding RNAs, than other domains (Supplementary Fig. 3f-g). Taken together, these results demonstrate that TIGs are enriched for developmental regulators and that their conserved boundaries are of potential regulatory importance.

**Single-gene domain boundary perturbation leads to dysregulation**. Next, we sought to investigate the effect of single-gene domain boundaries disruption on the gene they insulate throughout ES cell differentiation. We refined the list of developmental regulators curated from HUES64 cells to specifically include early developmental regulators (eDR), which displayed a stronger enrichment in both CTCF loop domains and TIGs than other developmental regulators (Supplementary Fig. 4a). The majority of these regulators are located within CTCF loop domains (89%, 33/37) (Fig. 4a), 25 of which are TIGs (Supplementary Fig. 4b-d), 8 are in multi-gene domains (Supplementary Fig. 4e), and 4 are located in CTCF loop domain-free regions (Supplementary Fig. 4f). To enable functional characterization of a representative TIG, we chose the *SOX17* locus as it is isolated by strong boundaries (boundary interaction strength adjusted *P*-value = 3.5e-9 based on Hi-C data) and encodes a member of the SOX (SRY-related HMG-box) family of transcription factors that is specifically induced in early endoderm differentiation by distal enhancers[30] (Fig. 4a, b and Supplementary Fig. 2a).

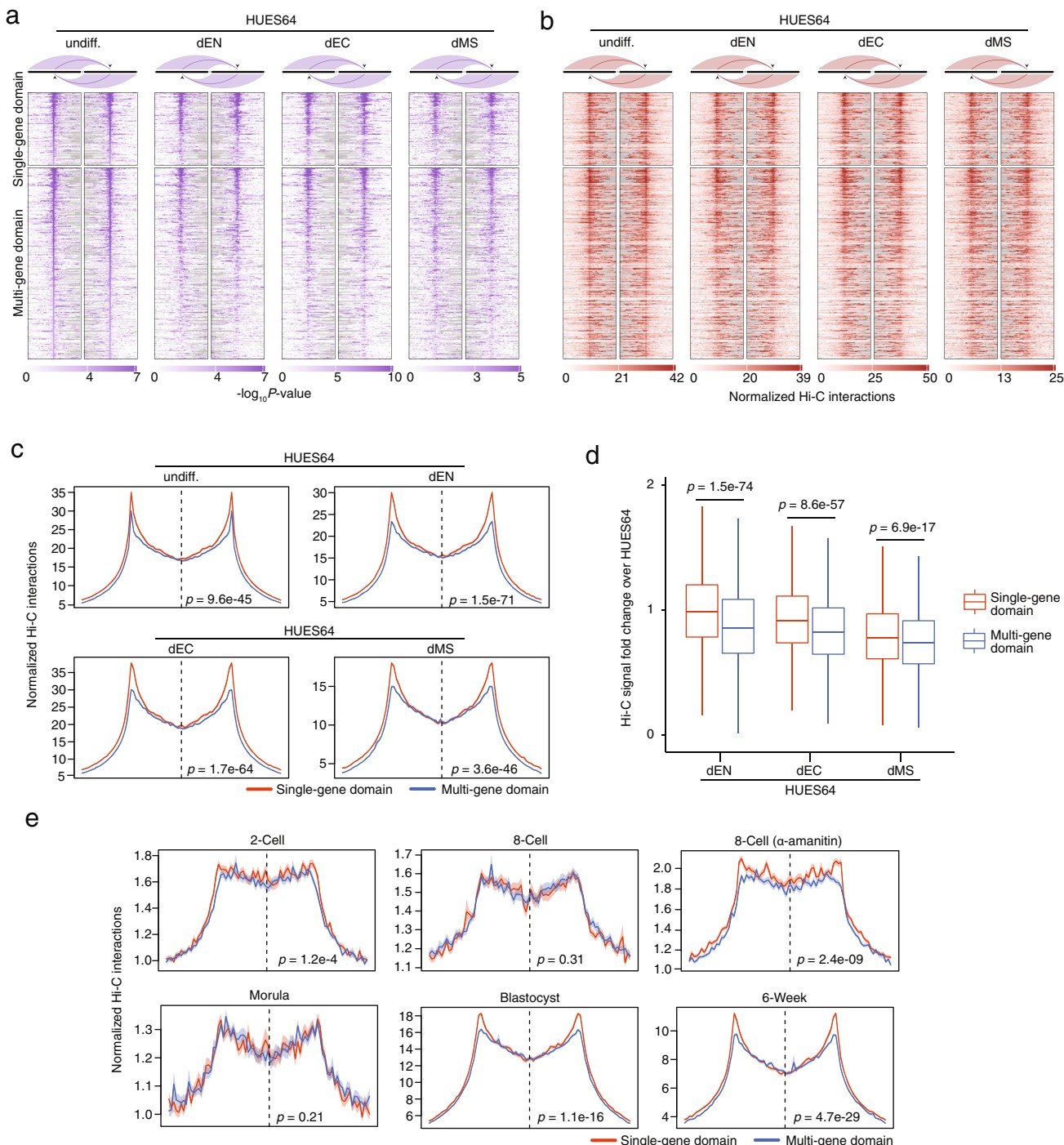

We designed two sgRNAs flanking the 5' centromeric boundary of the *SOX17* CTCF loop domain (Boundary 2), which is about 300 kb away from the locus (Fig. 4b and Supplementary Fig. 5a), and derived three independent homozygous SOX17$^{Δ5'CTCF}$ clones in the female iPSC line ZIP13K2[31]. The switch to an iPSC line has various practical benefits, such as sharing material and data across labs, and Hi-C data for human ESCs and iPSCs are very similar[32]. We further confirmed that the CTCF occupancy and Hi-C boundary strength at the SOX17 locus are also similar between human ESCs and iPSCs (Supplementary Fig. 6a).

We then confirmed the deletion of a 5 kb region that includes two CTCF peaks (Fig. 4c, Supplementary Fig. 5a, b) and the corresponding boundary interaction loss (Boundary 2) along with an increased interaction specifically in definitive endoderm at the upstream boundary (Boundary 1) in one of the boundary deletion cell lines (SOX17$^{Δ5'CTCF#8.2}$) (Fig. 4d, e, Supplementary Fig. 5a, b and Supplementary Data 3). We also identified a significant reduction of intraloop domain contacts (*SOX17* loop domain, *SOX17* upstream loop domain) in SOX17$^{Δ5'CTCF#8.2}$ iPSCs as well as a reduction of endoderm-specific enhancer contacts between the *SOX17* promoter and its most distal regulatory element DRE (A)) (Fig. 4d, e and Supplementary Fig. 6b, c). In concordance with highest intergroup contact correlations (Supplementary Fig. 6d) and without any further evidence of ectopic enhancer adoption or alternative enhancer

**Fig. 2 CTCF loop domain boundaries in ES cell differentiation. a** Boundary-anchored virtual 4 C heatmap of the domain boundaries in HUES64 and its derivatives. The locations of domain boundaries were identified in HUES64 Hi-C data. The −log10 *P*-value (before adjusting for multiple comparisons) obtained from Fit-Hi-C software are shown. Each row represents the domain of one gene. The strongest domain (i.e., that with the lowest Hi-C interaction *P*-value between boundaries) per gene is shown if there are multiple domains for that gene. **b** Boundary-anchored virtual 4 C heatmap of the domain boundaries identified from HUES64 Hi-C data plotted by using the Hi-C data of HUES64 and its derivatives as the underlying contact maps. The normalized Hi-C interactions are shown. The ordering is the same as in **a**. **c** Boundary-anchored virtual 4 C average profile of the domain boundaries in HUES64 and its derivatives. The locations of domain boundaries were identified in HUES64 Hi-C data. The normalized Hi-C interactions are shown. The dotted line separates the left and right boundary regions, which represent the regions in the left and right heatmap in **b**. Average signals across all boundaries are shown with the shaded area indicating the standard error. Two-sided Wilcoxon test was used to determine significance level of boundary-to-boundary interactions between the two groups. Data are presented as mean values ± SE. **d** Hi-C signal fold-change of boundary-to-boundary interactions in HUES64 derivatives over HUES64 cells. Two-sided Wilcoxon test *p*-value is shown. Hi-C signals were normalized by library size in individual samples prior to the analysis. The box indicates the interquartile range (IQR), the line inside the box shows the median, and whiskers show the locations of either 1.5 × IQR above the third quartile or 1.5 × IQR below the first quartile, $n = 3,310$ boundary-to-boundary interactions for single-gene domain, $n = 8,729$ boundary-to-boundary interactions for multi-gene domain. **e** Boundary-anchored virtual 4 C average profile of the domain boundaries at the 2-cell, 8-cell, 8-cell treated with α-amanitin, morula, blastocyst stages, and 6-week embryos. The locations of domain boundaries were identified in HUES64 Hi-C data. CTCF expression is inhibited under α-amanitin treatment at the 8-cell stage. The normalized Hi-C interactions are shown. The dotted line separates the left and right boundary regions. Average signals across all boundaries are shown with the shaded area indicating the standard error. Two-sided Wilcoxon test was used to determine the significance level of boundary-to-boundary interactions between the two groups. Data are presented as mean values ± SE.

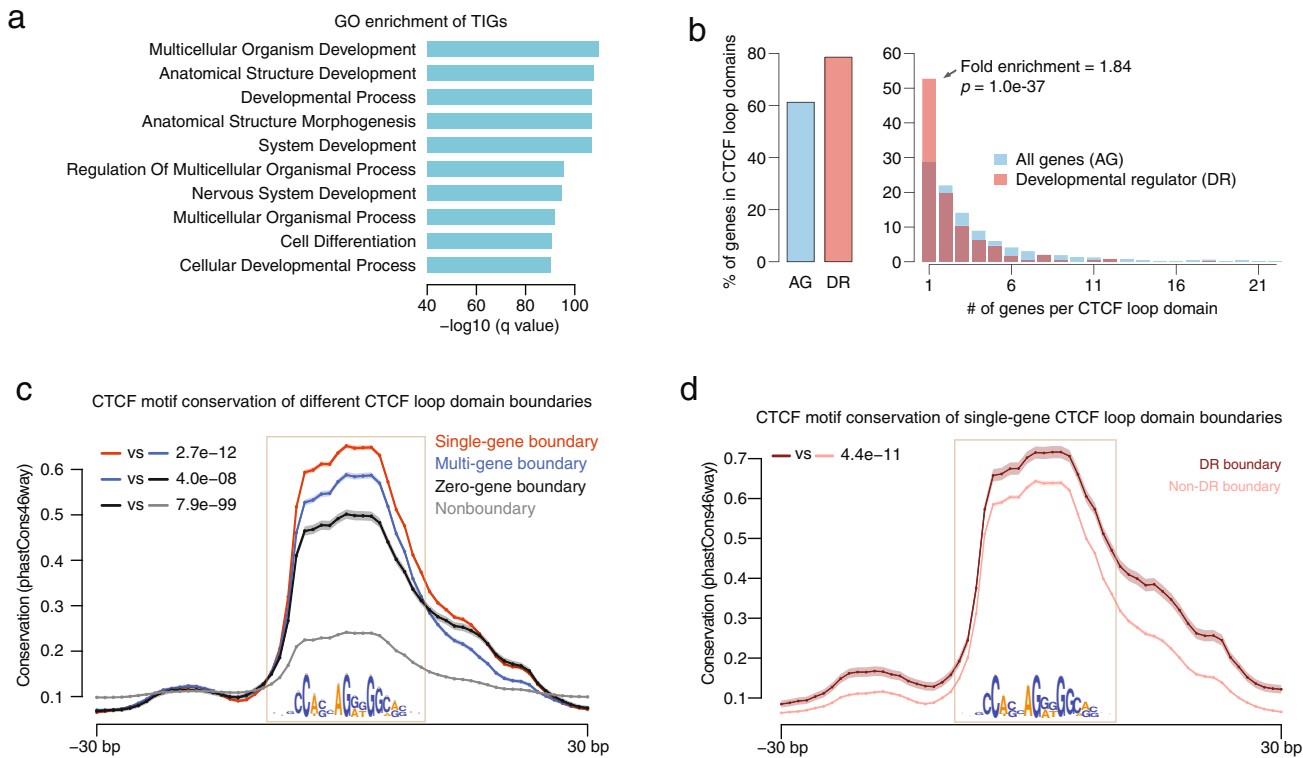

**Fig. 3 Topological insulation of developmental regulators. a** Gene Ontology (GO) enrichment of TIGs in HUES64 shows developmental processes as top terms. **b** Enrichment of developmental regulators in single-gene CTCF loop domains. The left panel represents the percentage of all genes (AG) or developmental regulators (DR) located within domains. The right panel represents the percentage of AG or DR located within domains containing an increasing number of protein-coding genes. *P*-values calculated by two-sided Fisher's exact test. **c** Evolutionary conservation of consensus CTCF motifs at boundaries of single-gene, multi-gene, and zero-gene CTCF loop domains and nonboundaries. Nonboundary CTCF motifs represent the motifs that are outside of any domain boundaries. The motif region is shown in the box and the motif sequence is displayed. The average conservation score across placental mammals across all boundary regions is shown. The shaded area indicates the standard error. Two-sided Wilcoxon test *p*-value tested in the motif regions is shown. Data are presented as mean values ± SE. **d** Evolutionary conservation of consensus CTCF motifs at DR boundaries and non-DR boundaries. See more descriptions in **c**. Data are presented as mean values ± SE.

looping due to boundary perturbation (Fig. 4d, e), we concluded that there was a decreased frequency of enhancer-gene contacts during definitive endoderm formation between the *SOX17* promoter and its tissue-specific enhancer DRE (A)) in the SOX17[Δ5'CTCF#8.2] cell line (Fig. 4f).

SOX17 is known to be a key early endoderm transcription factor[33,34] and is frequently used to identify embryonic

endodermal tissues, e.g., primitive, visceral, and definitive endoderm[30]. Together with the transmembrane C-X-C chemokine receptor 4 (CXCR4), SOX17 is used to specifically confirm definitive endoderm cell identity[30,34]. Notably, when we used directed differentiation conditions for generating definitive endoderm, we found a strong reduction in SOX17[+]/CXCR4[+] cell populations in all SOX17[Δ5'CTCF] isogenic clones (on average

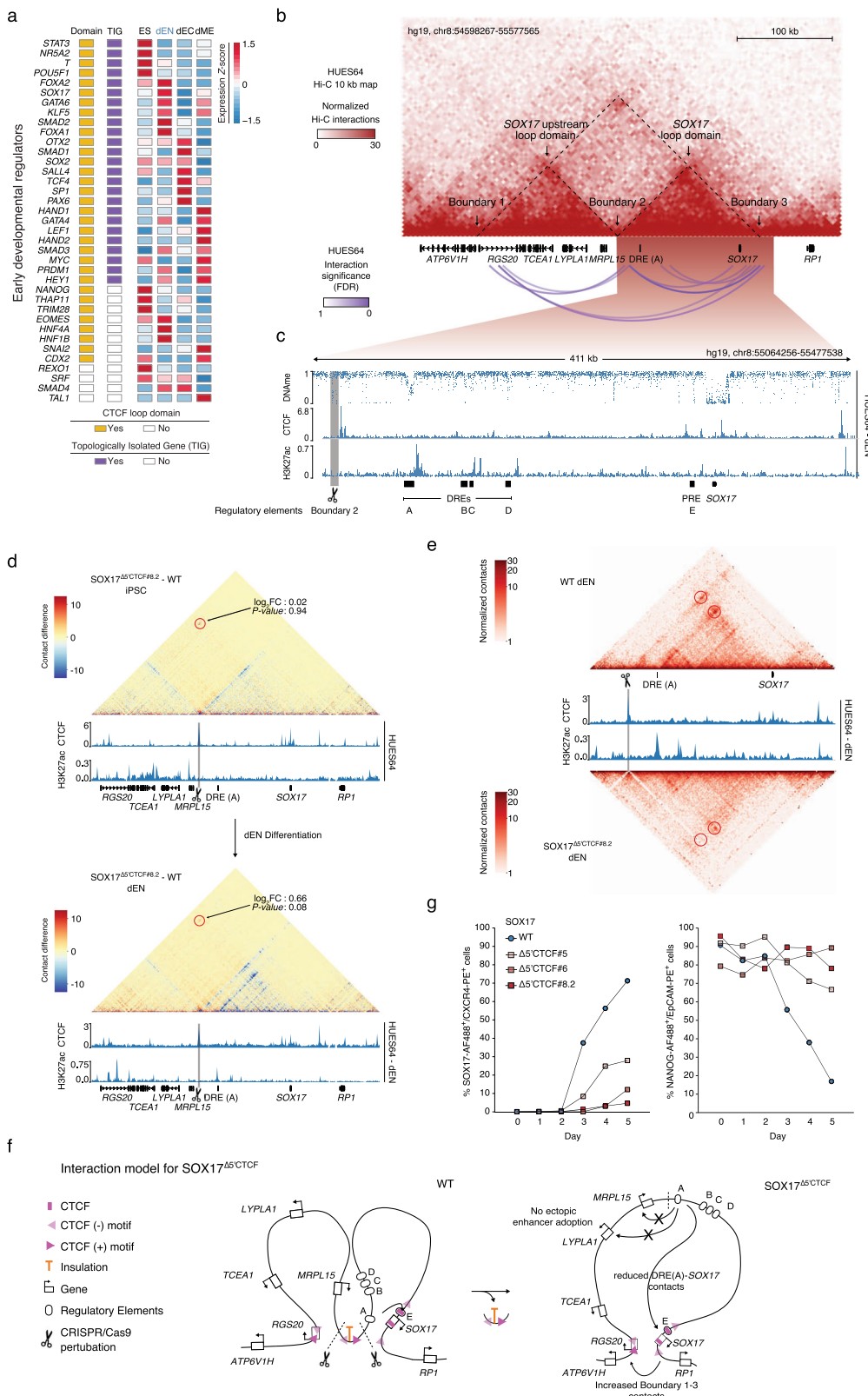

4.68–27.95%) compared to wild-type cells (71.3%) (Fig. 4g and Supplementary Fig. 5c, d). To assess whether the mutant cells have already exited pluripotency and lost their epithelial character due to epithelial-to-mesenchymal-transition (EMT)[30,35], we utilized the pluripotency transcription factor NANOG in combination

with the transmembrane glycoprotein Epithelial Cell Adhesion Molecule (Ep-CAM). We found reduced NANOG+/ Ep-CAM+ cell populations only in wild-type cells (16.9%) while population numbers remained comparably high over time in SOX17$^{\Delta5'CTCF}$ (on average 66.7–89.25%, Fig. 4g and

**Fig. 4 Single-gene versus multi-gene domain boundary perturbation highlights TIG-dependent gene regulation. a** Heatmap of early developmental regulators displays information on CTCF loop domains, TIGs, and expression in the embryonic stem cell differentiation process. The RPKM (Reads Per Kilobase per Million mapped reads) value of gene expression in embryonic stem (ES) cells, definitive endoderm (dEN), ectoderm (dEC), and mesoderm (dME) were row Z-scored. **b** Multi-layered display of the *SOX17* locus as a representative TIG at chr8:54598267-55577565. HUES64 CTCF loop domains are displayed as arcs below a normalized Hi-C interaction map. **c** Multi-layered display of HUES64 derived dEN WGBS, CTCF, and H3K27ac ChIP-seq profiles. Putative *SOX17* distal regulatory elements (DRE) and proximal regulatory elements (PRE) are highlighted in black bars and given capital letters. The deleted centromeric *SOX17* boundary (Boundary 2) is highlighted in grey marked by a scissor. **d** Capture Hi-C subtraction maps in iPSCs (upper panel) and dEN cells (lower panel) at the *SOX17* locus. The relative contact difference between the two samples (SOX17$^{\Delta5'CTCF\#8.2}$wild-type) in either iPSCs or dEN cells are shown on top of HUES64 or HUES64 derived dEN CTCF and H3K27ac ChIP-seq profiles. Boundary 1 + 3 contact quantifications are highlighted in red circles. The deleted centromeric *SOX17* boundary (Boundary 2) is highlighted in grey marked by a scissor. *SOX17* DRE (A) and gene bodies are highlighted in black bars. **e** Capture Hi-C maps in dEN wild-type (upper panel) and SOX17$^{\Delta5'CTCF\#8.2}$ (lower panel) at the *SOX17* loop domain. The normalized capture Hi-C contact maps are overlaid with HUES64 derived dEN CTCF and H3K27ac ChIP-seq profiles. Relative contact differences between Boundary 2+3 or between the *SOX17* promoter and DRE (A) are highlighted in red circles. The deleted centromeric *SOX17* boundary (Boundary 2) is highlighted in grey and marked by a scissor. Putative *SOX17* DRE (A) and *SOX17* gene body are shown as black bars. **f** Simplified 2D-model of the *SOX17* boundary 2 perturbation in wild-type or SOX17$^{\Delta5'CTCF}$ cells. Genes are depicted as white rectangles, regulatory elements as white ellipses. Crucial boundary related CTCF-ChIP-seq peaks are shown in pink; available motif-orientations are highlighted as arrows. Insulation is shown in orange. Dashed lines and scissors indicate the predicted Cas9 cut sites at boundary 2. **g** Fluorescence activated cell sorting (FACS) time-course data of wild-type and SOX17$^{\Delta5'CTCF}$ iPSC during directed differentiation towards definitive endoderm. SOX17 and CXCR4 (CD184) are depicted as percentage SOX17$^+$/CXCR4$^+$ in bulk cell populations. Corresponding NANOG and Ep-CAM (CD326) are depicted as percentage NANOG$^+$/Ep-CAM$^+$ in bulk cell populations ($n = 2$ biologically replicates). Data are presented as mean values.

Supplementary Fig. 5c, d), suggesting a boundary-dependent deregulation of SOX17 gene control and a failure to properly exit pluripotency.

To compare the effect with a multi-gene domain boundary deletion, we chose the *NANOG* locus as it is also isolated by strong boundaries (FDR=5.38e-10) in Hi-C data and encodes a highly expressed pluripotency TF in human iPSCs (Fig. 4a and Supplementary Fig. 5e)[36]. Two sgRNAs flanking the 3' centromeric boundary of the *NANOG* CTCF loop domain were designed, about 20 kb away from the *NANOG* locus (Supplementary Fig. 5e, f). We derived one homozygous and two heterozygous NANOG$^{\Delta3'CTCF}$ isogenic clones in the female iPSC line ZIP13K2[31], in which deletions of a 2 kb region including one CTCF motif were further confirmed (Supplementary Fig. 5f, g). Interestingly, we did not observe deregulation of relative NANOG protein and mRNA levels in NANOG$^{\Delta3'CTCF\#21}$ compared to wild-type cells (Supplementary Fig. 5h, i). mRNA expression levels of all other genes localized within the *NANOG* CTCF loop domain were also not found to be altered in NANOG$^{\Delta3'CTCF\#21}$ compared to wild-type cells (Supplementary Fig. 5i). Thus, we identified an important role for single-gene domain boundary-dependent regulation of genes as indicated by the *SOX17* locus and the fusion of both *SOX17* CTCF loop domains highlighted by a strongly disrupted differentiation outcome.

**SOX17 boundary perturbation leads to endoderm differentiation failure.** Next, we sought to confirm the absence of CTCF in SOX17$^{\Delta5'CTCF}$ cells and performed CTCF-ChIP qRT-PCR on *SOX17* Boundary 2 and control regions (Fig. 4b, c). We observed a genotype-specific loss of CTCF occupancy within and spanning *SOX17* Boundary 2 in SOX17$^{\Delta5'CTCF\#8.2}$ compared to wild-type iPSCs (Supplementary Figs. 5a and 7a). Due to the loss of this insulator in SOX17$^{\Delta5'CTCF\#8.2}$ cells, we next aimed to assure the absence of a potential adoption of the *SOX17* DRE (A) by upstream genes in an endoderm-specific context (Fig. 4b, c). Therefore, we isolated different CXCR4 subfractions for RNA-seq followed by differential gene-expression analysis; using this approach, we confirmed normal regulation of *SOX17* CTCF loop domain-associated genes in dEN, except for SOX17 (Supplementary Fig. 7b). Expression of SOX17 was exclusively observed in CXCR4$^+$ SOX17$^{\Delta5'CTCF\#8.2}$ cells, which comprised only a minor fraction of the population (on average 4.68–27.95%) (Fig. 4g and

Supplementary Fig. 7b). This data suggest a boundary-dependent deregulation of *SOX17* that is not associated with ectopic DRE-adoption by *SOX17* CTCF upstream loop domain-related genes.

To gain more insights into the transcriptomes of differentiated populations, we performed principle component analysis (PCA) of the 100 most variable genes across all samples (Fig. 5a and Supplementary Fig. 7c-g). Interestingly, SOX17$^{\Delta5'CTCF\#8.2}$ CXCR4$^+$ and wild-type CXCR4$^-$ cell populations closely clustered together on an endodermal differentiation trajectory roughly between undifferentiated and CXCR4$^+$ wild-type populations (Fig. 5a). Since CXCR4$^+$ wild-type and CXCR4$^-$ SOX17$^{\Delta5'CTCF\#8.2}$ populations comprised the respective majority, we analyzed differentially expressed genes ($n = 1,506$) using GSEA for biological processes ($\log_2$FC > 2, $q$-value < 0.05) and found genes enriched for DNA replication and cell cycle checkpoint in CXCR4$^-$ SOX17$^{\Delta5'CTCF\#8.2}$ cells (Supplementary Fig. 7d). As described previously, determination of endodermal cell fate propensity is closely connected to the cell cycle[37], suggesting that CXCR4$^-$ SOX17$^{\Delta5'CTCF\#8.2}$ cells are transcriptionally delayed and about to enter definitive endoderm. When comparing wild-type and SOX17$^{\Delta5'CTCF\#8.2}$ CXCR4$^+$ populations (437 genes), we found genes associated with negative regulation of growth to be enriched in SOX17$^{\Delta5'CTCF\#8.2}$ cells (Supplementary Fig. 7e). Interestingly, when comparing the more differentiated CXCR4$^+$ SOX17$^{\Delta5'CTCF\#8.2}$ population with their CXCR4$^-$ counterpart (635 differentially expressed genes), we found genes associated with gastrulation and stem cell differentiation (Supplementary Fig. 7f) enriched in CXCR4$^+$ cells, which again points towards a developmental delay of SOX17$^{\Delta5'CTCF\#8.2}$ CXCR4$^-$ cells and a compromised ability to generate proper definitive endoderm.

**DKK4 deregulation implicates a WNT signaling defect in SOX17$^{\Delta5'CTCF}$ cells.** Surprisingly, pathway GSEA led to the identification of enriched WNT signaling in SOX17$^{\Delta5'CTCF\#8.2}$ CXCR4$^+$ compared to wild-type CXCR4$^-$ cells (Fig. 5a and Supplementary Fig. 7g). To further perform paired gene enrichment analysis in an unbiased way, we performed expression z-score clustering of the most variable genes ($n = 4,151$) throughout all groups (Fig. 5b). We defined three clusters: the endoderm/gastrulation cluster (2,282 genes), WNT signaling cluster (569 genes), and self-renewal/pluripotency cluster (1,300

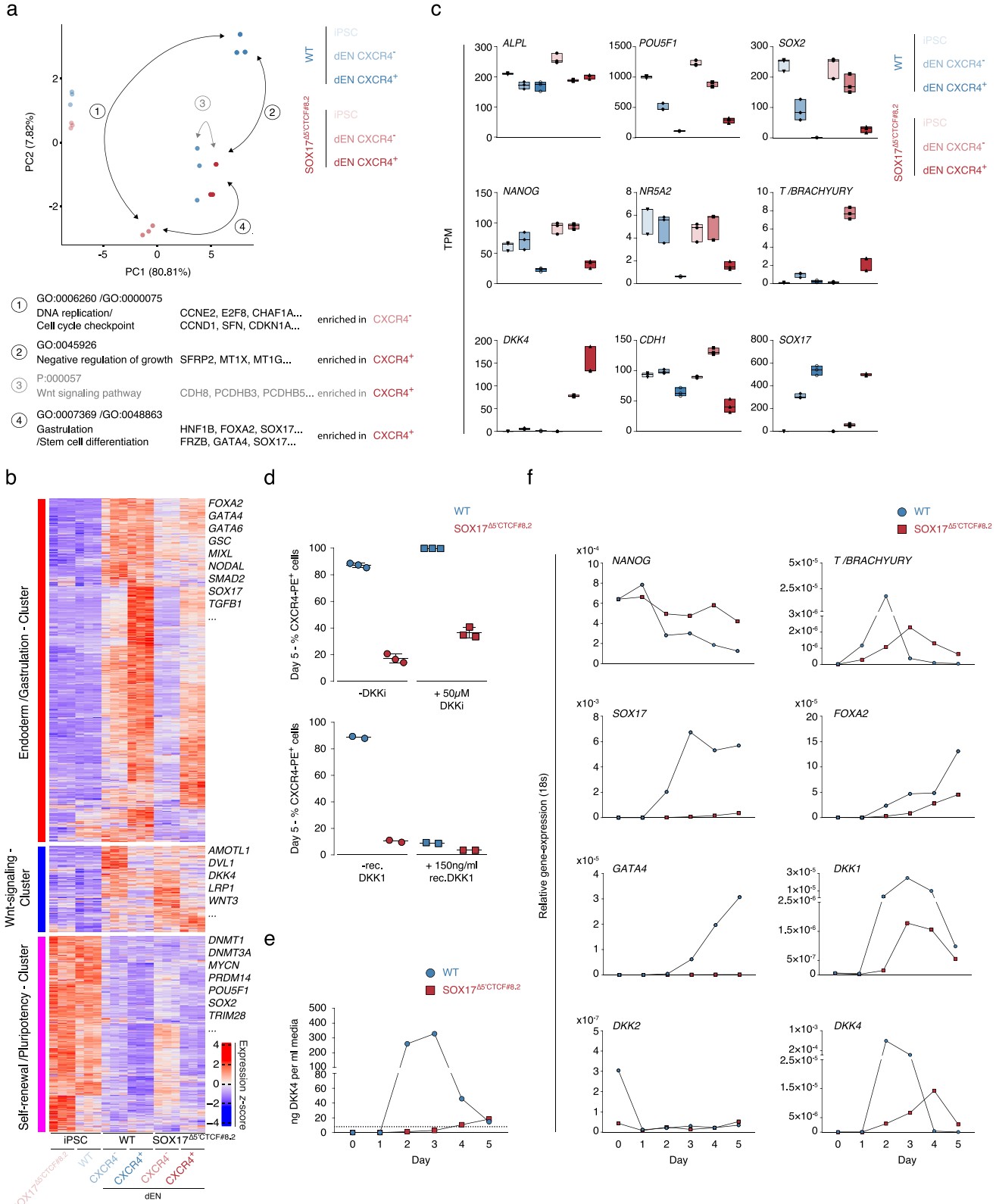

genes). Genes associated with the endoderm/gastrulation cluster were most highly expressed in CXCR4$^+$ wild-type cells while self-renewal/pluripotency cluster genes were found to be most highly expressed in both iPSC populations (Fig. 5b). Interestingly, WNT signaling-associated genes were most highly expressed in both CXCR4$^{-/+}$ SOX17$^{Δ5'CTCF#8.2}$ but also CXCR4$^-$ wild-type

populations, confirming our previous GSEA enrichment analysis (Fig. 5a, b). One of these marker genes, *DKK4*, was found to be exclusively upregulated in SOX17$^{Δ5'CTCF#8.2}$ cell populations (Fig. 5c). DKK4 has been shown to antagonize canonical WNT signaling by the inhibition of LRP5/6 interaction with WNT, forming a ternary complex with the transmembrane protein

**Fig. 5 SOX17 boundary perturbation leads to endoderm differentiation failure. a** Principal component analysis of RNA-seq data, depicting sample clusters by the use of the 100 most variable genes. The first two principal components (PCs) are displayed. Arrows and numbers indicate group comparisons. GSEA of differentially expressed genes between compared groups are indicated below; significantly enriched biological processes are depicted in black, pathways in gray. **b** TPM Z-score row normalized clustering of the most variable genes ($n = 4,151$). **c** TPM values shown for a subset of genes ($n = 3$ biological replicates). The box indicates the interquartile range (IQR), the line inside the box shows the median. **d** Wnt stimulation/antagonization utilizing fluorescence activated cell sorting (FACS) data of wild-type and SOX17$^{\Delta5'CTCF}$ iPSC at day 5 definitive endoderm. CXCR4 (CD184) is depicted as percentage CXCR4$^+$ in bulk cell populations. The upper panel shows DKK2/3/4 inhibition and controls (treatment for 5 consecutive days) ($n = 3$ biological replicates). Data are presented as mean values ± SD. The lower panel depicts human recombinant DKK1 treatment and controls (for 5 consecutive days) ($n = 2$ biological replicates). Data are presented as mean values. **e** DKK4 Enzyme-linked Immunosorbent Assay (ELISA), a quantitative measure of human DKK4 in cell culture supernatants over time ($n = 3$ biological replicates (averaged) over 2 experiments). Data are presented as mean values. **f** qRT-PCR of bulk populations from a subset of genes related to Wnt signaling, mesendoderm, endoderm, and pluripotency over 5 days endoderm differentiation. Expression values are depicted as relative gene-expression ($2^{-(\Delta Ct(GOI-18s))}$) ($n = 2$ biological replicates). Data are presented as mean values.

KREMEN that promotes internalization of LRP5/6[38]. Hence, expression of DKK4 may lead to insufficient canonical WNT signaling required for proper endodermal differentiation[39].

To explore the relevance of DKK4, we utilized a chemical inhibitor compound (9-Carboxy-3-(dimethyliminio)-6,7-dihydroxy-10-methyl-3H-phenoxazin-10-ium iodide), which led to partially rescued CXCR4$^+$ bulk population levels (on average 36.4%) (Fig. 5d). As a control experiment, we considered WNT inhibition instead by utilizing recombinant DKK1, which led to notably reduced CXCR4$^+$ bulk populations in wild-type (on average 8.95%) but not SOX17$^{\Delta5'CTCF\#8.2}$ cells (Fig. 5d). To test DKK4 levels released into the culture medium, we performed Enzyme-Linked Immunosorbent Assay (ELISA) over 5 days of dEN differentiation. We found a striking reduction of DKK4 levels in SOX17$^{\Delta5'CTCF\#8.2}$ culture supernatants compared to wild-type over time (Fig. 5e): DKK4 release was found to be slowly increasing and delayed over time, indicating no impact of WNT inhibition during differentiation, but rather being a consequence of deregulated *SOX17* gene control. The importance of WNT signaling, its role in endoderm and the functional relation between SOX17 and WNT signaling was demonstrated in studies utilizing *Xenopus* gastrulation[40,41]. Hence, we suggest a functional lack of SOX17$^{\Delta5'CTCF\#8.2}$ cells to respond properly to WNT signaling, most likely due to the boundary perturbation-dependent deregulation of *SOX17*.

In contrast, SOX17$^{\Delta5'CTCF}$ CXCR4$^-$ cell populations highly express mesendodermal markers, such as *T/BRACHYURY* and *NR5A2*, prematurely accompanied by high levels of pluripotent markers such as *NANOG*, *SOX2*, and *POU5F1* but not *ALPL1*. We also obtained elevated levels of the key epithelial marker *CDH1* as well as very low levels of *SOX17* to be expressed (Fig. 5c and Supplementary Fig. 7c). To test whether SOX17$^{\Delta5'CTCF\#8.2}$ cells may exert a rather delayed endoderm differentiation program, we performed qRT-PCR analysis on time-course differentiated bulk populations. We found strong expression reduction of the endodermal markers *SOX17*, *FOXA2*, and *GATA4* in concordance with stable *NANOG* expression over time, as confirmed by previous FACS data (Figs. 5f and 4g). We also found that the mesendodermal regulator *T/BRACHYURY* and WNT antagonists *DKK1/DKK4* showed reduced expression early in differentiation at day 1–3 and a clear expression onset delay of around 1–2 days, resulting in elevated expression levels at day 5 (Fig. 5f).

To investigate whether the observed phenotype is reversible by restoring the endoderm-required SOX17 expression levels, we made use of a destabilized ectopic SOX17-TagBFP expression system, which we randomly integrated into the SOX17$^{\Delta5'CTCF\#8}$ genetic background using the Piggy-BAC transposase (Supplementary Fig. 8a). Hygromycin-selected and TagBFP$^-$-sorted SOX17$^{\Delta5'CTCF\#8}$ cells were further cultured in bulk and named SOX17$^{DDSOX17}$. Endoderm differentiating SOX17$^{DDSOX17}$ cells were either treated

(SOX17$^{DDSOX17+}$) or not treated (SOX17$^{DDSOX17-}$) with a small molecule[42] named Shield-1 from day 2 onwards to either reverse the constitutive ectopic SOX17-TagBFP degradation or not, (Supplementary Fig. 8b). To first explore leakiness of our system, we performed a western blot assay, which revealed some minimal ectopic SOX17-TagBFP degradation in SOX17$^{DDSOX17-}$ undifferentiated iPSCs but also terminally differentiated dEN cells (Supplementary Fig. 8b). Interestingly, we found not only elevated levels of ectopic but also endogenous SOX17 protein in day 5 differentiated SOX17$^{DDSOX17+}$ cells compared to SOX17$^{DDSOX17-}$ cells, indicating a coupled activation of the endogenous *SOX17* locus (Supplementary Fig. 8b). From day 2 of endoderm differentiation onwards, we observed CXCR4$^+$ fractions to be restored to almost wild-type levels in SOX17$^{DDSOX17+}$ (Supplementary Fig. 8d). Surprisingly, we even found increased CXCR4$^+$ populations in SOX17$^{DDSOX17-}$ cells compared to the original knockout SOX17$^{\Delta5'CTCF\#8}$ cells, again indicating leakiness of our expression system (Supplementary Fig. 8d). Although SOX17$^{DDSOX17-}$ cells were found to be leaky for ectopic SOX17-TagBFP degradation, we observed a retained NANOG-expressing fraction of cells compared to SOX17$^{DDSOX17+}$ by immunofluorescence staining (Supplementary Fig. 8e).

Finally, to explore if the extent of transcriptional rescue in SOX17$^{DDSOX17+}$ and CXCR4$^{+/-}$ cells would resemble wild-type gene expression, we performed PCA of the 100 most variable genes across wild-type, SOX17$^{\Delta5'CTCF\#8}$ and SOX17$^{DDSOX17+}$ cells and found both populations including the undifferentiated iPSCs closely clustering with their wild-type matching cell populations (Supplementary Fig. 8f). In sum, our results suggest that SOX17$^{\Delta5'CTCF}$ cells can still exit pluripotency, but are delayed and trapped in a mesendoderm-like state due to WNT signaling nonresponsiveness via deregulated SOX17, leading to the eventual endoderm differentiation failure reversible by ectopic SOX17 expression.

**Constitutive CTCF loop domains and their essential functional roles.** After exploring the functional relevance of the boundaries of *SOX17* and *NANOG*, we aimed to more generally explore the function of boundaries genome-wide. As previously reported, CTCF-CTCF loops are preserved in different cell types[5,10,12,43], and constitutive loops are functionally important in tumors[10]. Therefore, we analyzed a collection of high-resolution Hi-C data obtained from 16 different cell lines and defined constitutive CTCF loop domains as those that were detected in at least 50% of all samples (Fig. 6a, b and Supplementary Data 4-5). We observed significant enrichment of early developmental regulators (18/37; fold enrichment = 5.24; *p*-value = 7.67e-10) among single-gene constitutive CTCF loop domains, with *SOX17*, *SMAD2*, *GATA4*, *STAT3*, *LEF1*, *FOXA2*, and *KLF5* as top representatives (Supplementary Fig. 9a). As expected, we found that the CTCF boundaries of constitutive CTCF loop domains are more conserved than those identified from individual cell types (Fig. 6c);

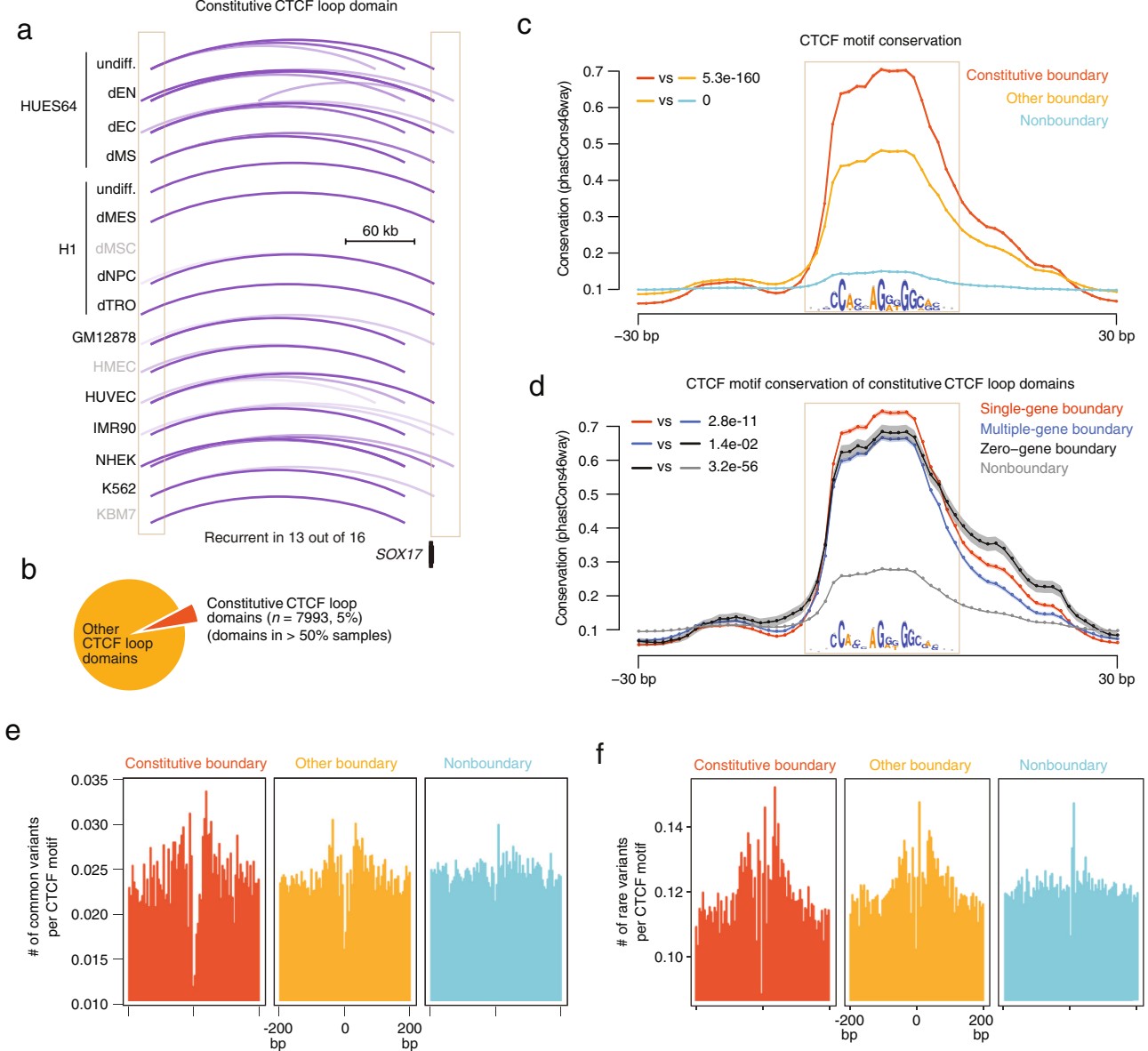

**Fig. 6 A functionally essential role of constitutive CTCF loop domains. a** Display of a constitutive domain at the SOX17 locus that is conserved across 13 out of 16 human cell types. Arcs represent the domains in different cell types. The constitutive boundaries are shown via the box. **b** The pie plot shows the proportion of constitutive domains and domains. Constitutive domains are present in >50% of samples, while domains are present in ≤50% of samples. **c** Evolutionary conservation of consensus CTCF motifs at constitutive domain boundaries, domain boundaries, and nonboundaries. Nonboundary CTCF motifs are those motifs not in any domain boundaries. The motif region is shown in the box and the motif sequence is displayed. The average conservation score across placental mammals across all boundary regions is shown. The shaded area indicates the standard error. Two-sided Wilcoxon test *P*-values tested in the motif regions are shown. Data are presented as mean values ± SE. **d** Evolutionary conservation of consensus CTCF motifs at single-gene, multi-gene, and zero-gene boundaries of constitutive domains and nonboundaries. Nonboundary CTCF motifs are the motifs not in any constitutive domain boundaries. See more descriptions in **c**. Data are presented as mean values ± SE. **e** The number of common variants per consensus CTCF motif site at constitutive domain boundaries, domain boundaries and nonboundaries. Common variants are genetic variants with an allele frequency larger than 1% in the 1000 genome project phase 3 data. **f** The number of rare variants per consensus CTCF motifs at constitutive domain boundaries, domain boundaries, and nonboundaries. Rare variants are genetic variants with allele frequency less than 1% in 1000 genome phase 3 data.

furthermore, TIG boundaries within-constitutive domains were more conserved than others (Fig. 6d). We therefore hypothesized that, if domain boundaries are essential for controlling the expression of the developmental regulators they contain, the disruption of such elements should be negatively selected for in individuals. To test this hypothesis, we analyzed the 1000 Genome Project phase 3 data, which contains 84.4 million variants identified from data on 2,504 individuals from 26 human populations, to interrogate frequent variants in CTCF loop domain boundaries. We observed a depletion of common variants (allele frequency >1% in the population) in domain boundaries relative to flanking regions, especially for constitutive CTCF loop domain boundaries (Fig. 6e); nonboundary CTCF sites did not demonstrate such a depletion. Rare variants (allele frequency <1% in the population) showed a similar but less pronounced trend (Fig. 6f). These findings indicate that boundaries of constitutive domains, if altered, are subject to purifying selection, thus supporting their essential role in preventing deleterious phenotypes.

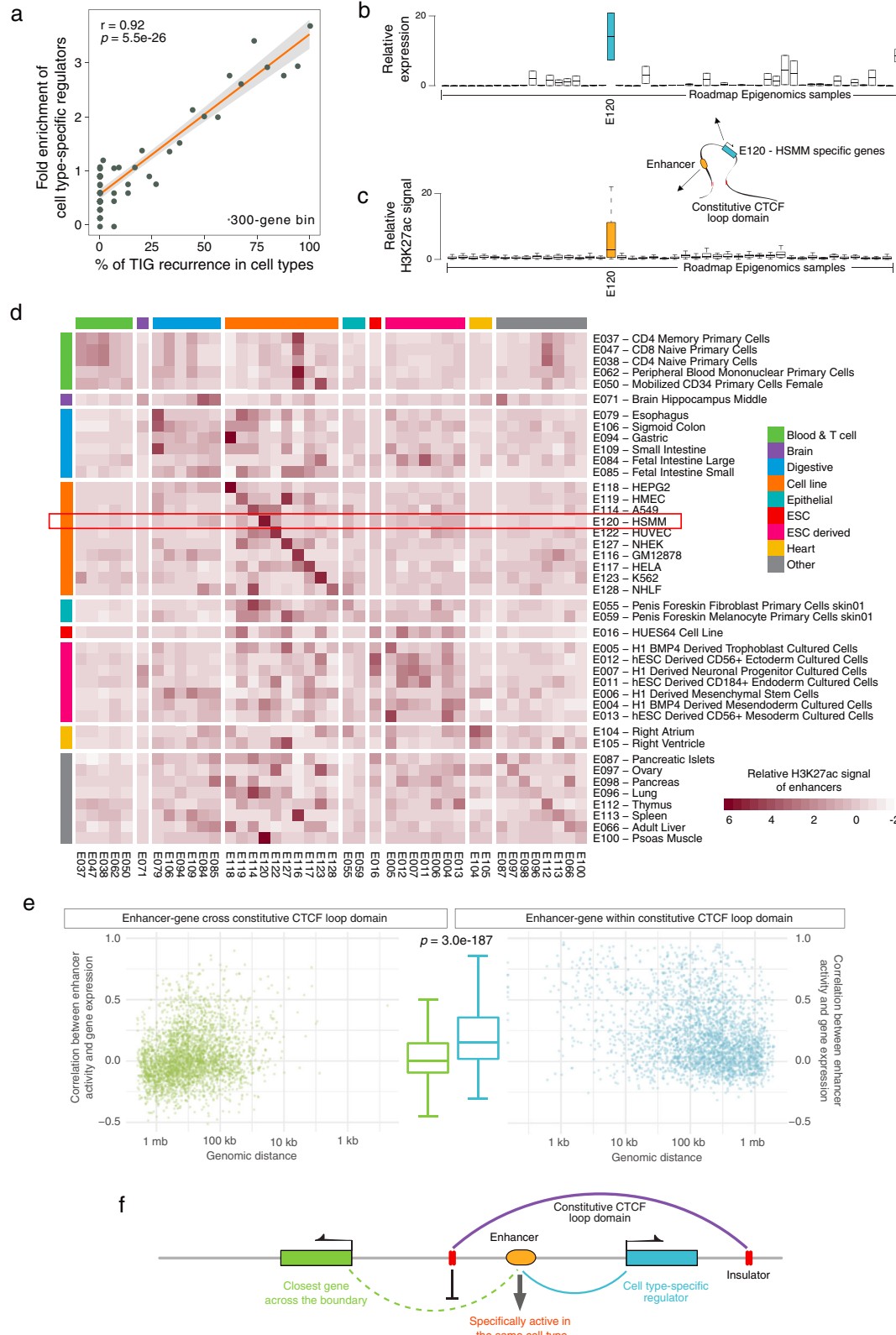

**Co-activation within and insulation across constitutive CTCF loop domains.** To further explore how these boundaries of constitutive domains influence the expression of genes they contain, we utilized the Roadmap Epigenomics Project dataset, which includes measurements of both gene expression (RNA-seq) and enhancer activity (H3K27ac ChIP-seq) in the same set of cells. First, we defined cell-type-specific regulators by selecting transcription factors that demonstrate cell-type-specific expression patterns across 54 tissues and cell lines (Methods). We observed that TIG recurrence in cell lines was positively correlated with enrichment of cell-type-specific regulators (Fig. 7a, r = 0.92, *P*-value = 5.5e-26). For consistency, we identified

**Fig. 7 Co-activation within and insulation across constitutive CTCF loop domains. a** Fold enrichment of cell-type-specific regulators against the recurrence of TIGs in multiple cell types. Each point represents 300 genes. Linear regression line with 95% confidence interval in light gray is shown. *P*-values calculated by two-sided Pearson correlation test. **b** Relative expression of cell-type-specific regulators in E120 across all cell types. Relative expression is the expression within a cell type normalized by the mean expression across all cell types. The box indicates the interquartile range (IQR), the line inside the box shows the median, and whiskers show the locations of either 1.5 × IQR above the third quartile or 1.5 × IQR below the first quartile, $n = 2$ genes. **c** Relative H3K27ac abundance of enhancers located within the same constitutive domain of the cell-type-specific regulators in E120 across all cell types. Relative H3K27ac abundance is the H3K27ac signal over mean H3K27ac signal across all cell types, which represents enhancer activity. The box indicates the interquartile range (IQR), the line inside the box shows the median, and whiskers show the locations of either 1.5 × IQR above the third quartile or 1.5 × IQR below the first quartile, $n = 28$ enhancers. **d** Mean H3K27ac abundance of enhancers located within the same constitutive domain of the cell-type-specific regulators. Each row depicts the mean of the relative H3K27ac abundance of enhancers located within the same constitutive domain of the cell-type-specific regulators in the corresponding cell type across all cell types. The red box indicates the mean value of the boxplot in **c**. Note that some tissues and cell lines are functionally related, which may drive the enrichment off the diagonal, such as hematopoietic cell-type-specific enhancers being also enriched in GM12878 and thymus. Enhancers specific to cell lines are stronger and uniquely enriched in a specific cell type, such as HEPG2 enhancers. Some enhancers show tissue-type-specific properties instead of cell type specificity, such as enhancers of ESCs and derivatives, hematopoietic cells, and heart cells. **e** Correlation between enhancer activity and within-constitutive domain-neighbor gene expression (left plot in green) and correlation between enhancer activity and cross-constitutive domain-neighbor gene expression (right plot in blue). The boxplot shows the whole distributions of the data shown on the left and right. The box indicates the IQR, the line inside the box shows the median and whiskers show the locations of 1.5 × IQR above the third quartile and 1.5 × IQR below the first quartile, $n = 3,611$ enhancers. Note that only cell-type-specific regulators (TIGs) and their constitutive domains are included in this analysis. **f** Model of co-activation of enhancer and cell-type-specific regulators within the same single-gene constitutive domain. The schematic also depicts the insulation function of constitutive domain boundaries.

constitutive TIGs and investigated their overlap with cell-type-specific regulators (Supplementary Fig. 9b). We found that those genes and enhancers localized within the same constitutive domain were co-activated in either a cell-specific or a tissue-specific manner (Fig. 7b-d and Supplementary Fig. 10). Further analyses demonstrated that enhancer activity is more correlated with its neighbor gene expression within-constitutive domains than outside of them (Fig. 7e, Wilcoxon test *P*-value < 0.0001). These results support the presumed insulation function of constitutive domains by restricting the enhancer activity to the targeted genomic region, and further demonstrate that topological isolation can add to precise control of local gene expression (Fig. 7f).

## Discussion

Elucidating the relationship between chromosome structure and gene regulatory programs is of broad interest. Increasing evidence has demonstrated that mediator cohesin loops, mostly enabling enhancer-promoter interactions, have substantial effects on gene regulation in diverse systems[7–9,20,43–46]. Similarly, CTCF cohesin loops, mostly functioning as insulators, have been proposed to constrain enhancer-promoter interactions for proper gene-expression patterns[5,10,12,14,47]. Here, we describe an aspect of genome organization that may facilitate the precise temporal and spatial control of key developmental regulators. Our model is supported by functional data showing that disruption of a CTCF loop domain boundary can strongly impact the lineage commitment of pluripotent cells. Integrative and systematic analyses further highlight sequence conservation, germline variant and boundary constraint profiles, which extend our functional case study and demonstrate the importance of CTCF loop domain boundaries on a genome-wide scale. This understanding of topological isolation and the precise control it may exert on developmental processes suggests a potential mechanism of co-evolution between transcriptional control and chromosome structure formation. Our work supports a model suggesting that gene duplication and its subsequent organization in its own single-gene domain may be a frequent way to evolve and acquire new gene functions without disrupting neighboring genes and their regulatory elements.

A previous study demonstrated that knockouts of CTCF loop domain boundaries leads to altered expression of nearby genes in

mouse ES cells, providing evidence that the maintenance of topological boundaries is important for the proper expression of these genes[5]. However, there is limited knowledge how boundary perturbation-induced gene-expression changes influence ES cell differentiation in humans. Our functional study on the *SOX17* locus demonstrated that disruption of CTCF loop domain boundaries strongly impacts cell lineage commitment of human iPS cells. In addition, enhancer adoption/hijacking after boundary perturbation has been widely observed and extensively studied in both development and tumorigenesis[10,14,29,47]. Here, we shown that the boundary knockout at the *SOX17* locus did not induce enhancer adoption by other genes but causes loss of proper enhancer regulation of its endogenous targets. Our results suggest a dual function of topological insulation—the boundary interaction not only constrains enhancer activity within the domain, but also facilitates enhancer-promoter interaction by bringing them into physical proximity (Fig. 4g). This observation may also imply the existence of diverse mechanisms of topological insulation[17], which need to be further dissected.

We demonstrate more pronounced boundary conservation of CTCF loop domains containing a single gene than those containing multiple genes, suggesting a possibly more critical role of boundaries of single-gene than those of multi-gene domains. This observation is in-line with findings of our boundary disruption experiment at the *NANOG* locus, which is a multi-gene CTCF loop domain, in ES cells, where we did not observe any phenotypic change and no genes within the locus had significantly altered expression levels. Since *NANOG* is highly expressed and required for the maintenance of pluripotency in ES and iPS cells[36,48], these results suggest that gene-expression regulation in this context is independent of boundary disruption in pluripotent cells. These results provide a preliminary understanding of the difference between single-gene and multi-gene CTCF loop domains, with the limitation of the currently still relatively small number of reported examples[9,10,14,15,17,47,49]. Validation of our findings with additional CTCF loop domain boundary functions in different cell states is needed to arrive at a better understanding of how 3D topological structures control the gene-expression program in multi-cellular processes. One of the barriers preventing such a functional study is the lack of a uniform phenotype; for instance, cell viability may not be a good indicator for all cellular processes[50], and in some cases, boundary disruptions of a CTCF

loop domain have not led to immediate gene-expression changes[51] and clear phenotypes[49]. Thus, it might be important to consider the specific context and the exact point in time when topological insulation may exert its control on gene expression[52].

Our study has demonstrated that 90% of early developmental regulators, many of which are essential, are localized in CTCF loop domains; this high representation led us to hypothesize that these regulators need to be shielded from interference by neighboring regulatory elements or need their own elements within an isolated topological domain to be accessible. By contrast, there might be more flexibility for regulators acting in somatic cells, as deregulation of such factors may cause a developmental disorder or tumorigenesis but might not be immediately lethal. In both cases, the boundary alterations via DNA mutations or structural variations might be able to be used as diagnostic markers or therapeutic targets across multiple disease types. For instance, a previous study demonstrated that forced chromatin looping by tethering enhancers to repressed γ-globin genes reactivated their expression by overriding the endogenous topological structures[53]. Another study developed a light-activated system to conduct endogenous gene-expression control via dynamic induction of artificial chromosome loops[54]. These recent technologies provide promising therapeutic approaches to treat diseases caused by 3D topological alterations, potentially leading to the emergence of 3D therapeutics.

## Methods

**Parameters**. Default parameters were used, if not otherwise specified, for all software and pipelines utilized in this study.

**Hi-C sequencing**. Hi-C libraries were prepared following the protocol described in Rao et al.[7]. Briefly, one million cells were crosslinked with 1% formaldehyde for 10 min at room temperature and then quenched with 0.2 M glycine solution. Cells were lysed and nuclei permeabilized with 0.5% sodium dodecyl sulphate for 10 min at 62 °C. Chromatin was digested with 100 U of MboI restriction enzyme (New England Biolabs). Ends of the restriction fragments were blunted and labeled with a biotinylated nucleotide and then ligated. Nuclei were pelleted, proteins were digested with proteinase K and crosslinks were reversed by heating at 68 °C overnight. DNA was sheared in a Covaris focused ultrasonicator to average fragment length of 400 bp. Size-selected DNA was enriched for biotinylated ligation products through binding to T1 streptavidin beads (Thermo Fisher). Libraries were prepared for Illumina sequencing by performing the end repair, A-tailing and adapter ligation steps with DNA attached to the beads. Hi-C libraries were amplified directly off the beads and purified for subsequent Illumina sequencing with 100 paired-ends.

**Identification of CTCF loop domains from Hi-C data**. Raw Hi-C reads were mapped to the hg19 version of the human genome and preprocessed using the Hi-C-Pro pipeline[55] (version 2.11.0-beta) to obtain uniquely mapped deduplicated interactions. These interactions were then aggregated into 10 kb genomic bins and normalized using the caICB algorithm in HiCapp[56] (v1.0.0). The high-confidence (i.e., significant, q < 0.01) interactions in the genomic range of 30 kb–2 Mb were identified using the Fit-Hi-C python package[23] (v1.0.1). By mapping the anchors of high-confidence interactions to CTCF sites (the union of CTCF motifs and CTCF-ChIP-seq peaks in the corresponding sample), we obtained CTCF-CTCF loops. We observed that some samples with low sequence depth had very few identified loops, because small counts led to a low power for interactions to pass the significance cutoff. Therefore, we applied a hard cutoff to obtain the top 10,000 CTCF-CTCF loops in these samples based on previous evidence regarding the number of these loops per cell type[5,7,10]. Subsequently, loops close to each other were clustered and merged to reduce redundancy (see below). These merged loops generated the final set of CTCF loop domains. Note that summits of merged loops were identified based on the Hi-C interaction significance and were used instead of the merged loops themselves to increase the resolution of anchor points. The same procedure was performed by using three other Hi-C loop detection methods with the recommended parameter settings by the original references: HiCCUPS[25] (-m 500 -r 10000 -f.1), SIP[24] (-res 10000 -fdr 0.05), and Homer[26] (-res 10000 -window 50000). The CTCF loop domains were compared across different methods, and were compared to insulated neighborhoods identified by cohesin ChIA-PET data in primed human ES cells[20].

**Identification of topologically isolated genes (TIGs) from Hi-C data**. Protein-coding genes (PCGs) were extracted from the RefSeq annotation of the hg19

version of the human genome. The transcription start sites (TSSs) of PCGs were compared to the localization of CTCF loop domains to decide how many PCGs are located within each CTCF loop domain. CTCF loop domains containing one PCG were named single-gene CTCF loop domains; CTCF loop domains containing more than one PCG are referred to as multiple-gene CTCF loop domains; and CTCF loop domains containing no PCGs were named zero-gene CTCF loop domains. The PCGs in the single-gene CTCF loop domains are referred to as Topologically Isolated Genes (TIGs). The use of the phrase "isolated" is meant to represent that a gene is localized by itself in a CTCF loop domain. To classify genes into these categories, we used the TSS instead of the whole gene body because the promoter region represents the key transcriptional regulator of a gene; it is the promoter that requires tight regulation by insulation of chromosomal regions in order to prevent mis-regulation by nearby elements[5].

**Boundary-anchored virtual 4 C visualization of Hi-C data**. To visualize the boundary interactions of many CTCF loop domains in a Hi-C data, we used the boundary-anchored virtual 4 C plot. It's a simple way to visualize the interactions between one boundary to the surrounding regions of the other boundary. More specifically, the left heatmap shows the Hi-C interactions between the surrounding genomic regions of the left boundaries and the right boundaries; The right heatmap shows the Hi-C interactions between the surrounding genomic regions of the right boundaries and the left boundaries. Any Hi-C matrix-like scores derived from Hi-C data can be shown by using such plot, such as the normalized Hi-C interactions and the Fit-Hi-C p-values. Then, the heatmaps can be aggregated by the columns of the left and right heatmaps to generate a shaded line plot with the line represents the average signal across columns and the shaded area represents the standard deviation signal across columns. The shaded line plot could be used to visualize the difference between multiple groups of interactions as well as calculate statistics. The left and right heatmaps are plotted into single shaded line plot with a dotted vertical line to separate them.

**Identification of constitutive CTCF loop domains and TIGs from multiple Hi-C datasets**. We used the CTCF loop domains identified from 16 Hi-C datasets to obtain the union CTCF loop domains across cell types (Supplementary Data 4). One key step in CTCF loop domain calling is to use CTCF-binding sites to filter high-confidence interactions identified from Hi-C data, because Hi-C interactions may contain other types of chromosomal structures such as enhancer-gene loops that do not belong to CTCF cohesin loops. In practice, if particular CTCF peaks fail to be detected in the CTCF ChIP-seq data of one or several samples, even if Hi-C data were to show a high-confidence interaction loop at that genomic position, we would miss the CTCF-CTCF loop in these samples. To avoid this scenario, we used the same consensus CTCF-binding sites for each sample instead of the binding sites obtained from individual ChIP-seq data to identify CTCF loop domains in the constitutive CTCF loop domain analysis. The constitutive CTCF loop domains were then defined as those CTCF loop domains that were identified in at least 50% of all cell types (see above), and the genes located within single-gene constitutive CTCF loop domains are referred to as constitutive TIGs (cTIGs).

**Clustering and merging of redundant loops**. We designed a two-step iterative clustering algorithm to cluster and merge paired-end loops within a certain genomic range cutoff; here we used a 1 kb region of boundary overlaps. In the preclustering step, we ranked all loops by their chromosome position and subsequently divided them into two groups based on whether they had even or odd ranks. We then used the pairToPair command in bedtools[57] (v2.25.0) to investigate the overlaps of boundaries between any paired loops from the two sets. The loops in one set that overlapped with any loops in the other set were merged to form new loops with union boundary regions. The loops in one set having no overlaps with any loops in the other set were retained. The merged and retained loops were used as the input for the next iteration. We iteratively applied this process N = 50 times to obtain preclustered loops. In the complete-clustering step, we used the same strategy as in the previous step, except for searching for the overlaps between the preclustered loops and other loops in the same set, instead of dividing them into two loop sets. Self-pairs were excluded from the analysis. In this step, the iterations were continued until the algorithm converges and no paired-end loops can be merged anymore. This two-step procedure was able to cluster and merge a large number of redundant loops in any given genomic range cutoff in a short time period.

**Identification of CTCF motifs and their conservation across species**. CTCF motif loci and orientations in the hg19 version of the human genome were identified using FIMO[58] (v4.11.1). For this analysis, we used the consensus CTCF motif MA0139.1 from the JASPAR CORE 2016 vertebrates database[59]. Motif conservation information was obtained from the UCSC "phastCons46wayPlacental" track.

**Evolutionary analysis of human CTCF loop domain boundaries**. The CTCF motif coordinates of human CTCF loop domain boundaries were liftovered to 45 vertebrate genomes with parameter: -minMatch = 0.9. The motifs successfully liftovered were called present, otherwise absent, in the corresponding genome. The

 

percent of present motifs in different CTCF loop domain groups across species were studied.

**Identification of consensus CTCF-binding sites.** The CTCF ChIP-seq peaks in 142 different cell lines and tissues (Supplementary Data 5), which were identified using the same settings and contained at least 10,000 peaks, were downloaded from the Cistrome database[60]. The CTCF peaks (p < 1e-9, peak significance over input) detected in more than 30% of all unique cell types were defined as consensus CTCF-binding sites. The coordinates of ChIP-seq peaks were overlaid with CTCF motifs to obtain orientation information and highest resolution of CTCF-binding sites. Specifically, for the ChIP-seq peaks overlapping with CTCF motif(s), the motif coordinates and orientations were used instead of the peak coordinates. For the ChIP-seq peaks not overlapping with any CTCF motif, the peak coordinates were used and the orientations were set as 'unclear'.

**Clustered and typical CTCF-binding sites.** CTCF ChIP-seq data were analyzed in a similar way as the enhancer analysis of the ROSE pipeline[61]. Specifically, CTCF peaks were merged within a maximal distance of 12.5 kb. The merged peaks were ranked by increasing total ChIP-seq signal, and plotted against the total ChIP-seq signal. This plot showed a clear transition point in the distribution of CTCF occupancy where the total signal began increasing rapidly. The transition point was the x-axis point for which a line with a slope of 1 was tangent to the curve. We then defined peaks above this point to be clustered CTCF-binding sites, and peaks below that point to be typical CTCF-binding sites. Thus, clustered CTCF-binding sites represent those sites with broad and high CTCF occupancy, while typical CTCF-binding sites represent sites with narrow and low CTCF occupancy.

**Identification of cell-type-specific regulators.** The gene-expression matrix providing transcripts per million (TPM) for 57 samples was downloaded from the Roadmap Epigenomics project[62]. Sample E000, which represents the universal human reference, and three redundant samples (E056, E058 and E061) were removed from the analysis. The gene-expression matrix of the remaining 53 samples (Supplementary Data 5) was then used to identify cell type specifically expressed genes as previously described[63]. We employed the dataset of the Roadmap Epigenomics Project to identify cell-type-specific regulators because it contains diverse cell and tissue types such as stem cells, differentiated cells, primary cells, tissues and immortalized cell lines. The cell-type-specific regulators were defined as transcriptional regulators that were highly induced in certain cell types. Specifically, genes were selected that met the following criteria: (i) entropy less than 0.8; the entropy is calculated as: prop < -x/sum(x); Entropy = -sum(prop*log (prop), na.rm = T)/log(length(prop)), where x is the TPM vector across samples; and (ii) the maximal TPM across samples is larger than 10; in larger than 0 and less than or equal to 5 samples the gene has at least 7-fold higher expression than the average expression of the gene in all cell types.

**Identification of enhancers and analysis of their H3K27ac enrichment.** Enhancers were collected from both the Fantom5 database[64] and the Roadmap Epigenomics Project[62]. The enhancers from Fantom5 were directly downloaded from the website (http://slidebase.binf.ku.dk/human_enhancers/). The enhancers from the Roadmap Epigenomics project were identified from H3K27ac ChIP-seq data of 98 samples. The aligned reads from 98 samples were downloaded as bed files and converted to bam and bigwig files using MACS2[65] (v2.1.0.20150731) and bedtools[57] (v2.25.0). Narrow peaks (p < 1e-9, peak significance over input) in the same samples called from MACS2 were downloaded and used to identify enhancers via the ROSE pipeline[61] (v0.1). The enhancers from both databases were merged and the union sets were designated as the final list of enhancers. Average H3K27ac signals for enhancers were obtained from the ChIP-seq bigwig files and normalized to signals per 10 million reads for each library. Forty-five of 98 samples for which both H3K27ac ChIP-seq and RNA-seq data was available were used for the analysis (Supplementary Data 5). Enhancers with a maximal signal of less than 5 across all 45 samples were treated as inactive enhancers and removed from the analysis. We found that our results were robust when this cutoff was changed.

**Gene sets and enrichment analyses.** Developmental regulators were genes overlapping with transcription factors and genes under the GO term GO:0032502 - developmental processes. Early developmental regulators were obtained from Tsankov et al.[34]. Gene set enrichment analysis was performed using Fisher's exact test.

**Germline variants from 1000 Genome Project phase 3 data.** We analyzed the 1000 Genome Project phase 3 data, which contains 84.4 million variants identified from data on 2504 individuals from 26 populations, to interrogate frequent variants in CTCF loop domain boundaries. Common variants are genetic variants with an allele frequency larger than 1%; and rare variants are genetic variants with an allele frequency less than 1%.

**Data visualization.** Juicebox[66] was used to generate .hic files of Hi-C data to visualize in the WashU EpiGenome Browser[67] to create the genome track figures.

Other figures were plotted in the R environment (https://www.r-project.org) using basic plotting functions and packages of ggplot2[68], pheatmap (https://cran.r-project.org/web/packages/pheatmap/index.html). APA plots were generated by Juicebox[66].

**Molecular cloning of boundary knockout constructs.** For CRISPR/Cas9 mediated targeting of either *SOX17* or *NANOG* boundary knockout constructs we utilized pSpCas9(BB)-2A-GFP (PX458), which was a gift from Feng Zhang (Addgene plasmid # 48138; http://n2t.net/addgene:48138; RRID: Addgene_48138)[69]. Prior to small guide RNA (sgRNA) cloning, pX458 was initially modified and further renamed into 2X_pX458. 2X_pX458 harbors an additional independent U6-promoter followed by a small guide RNA (sgRNA) scaffold expression cassette, which allows the insertion of an additional sgRNA by SapI restriction enzyme cloning. To generate 2X_pX458, pX458 and the synthetized SapI sgRNA expression cassette (IDT, find sequence below) were digested with KpnI (New England Biolabs, R3142S). Next, the SapI sgRNA expression cassette was ligated into the KpnI linearized pX458 in a 3:1 molarity ratio using T4 DNA-ligase (New England Biolabs, M0202S) according to the manufacturer's instructions followed by transformation and Sanger sequencing to verify successful cloning.

sgRNA-cloning was performed with NEBuilder HiFi DNA Assembly Master Mix (New England Biolabs, E2621S) according to manufacturer's instructions using BbsI-linearization of 2X_pX458 for the first sgRNA and SapI linearization of 2X_pX458 for the second sgRNA as backbone, combined with single stranded oligonucleotides containing the sgRNA sequences as inserts (1:3 molar ratio) (Supplementary Table 1). Bacterial transformation and Sanger sequencing was performed to verify successful cloning. Empty 2X_pX458 was deposited on addgene.org under ID #172221. The 2X_pX458 derived *SOX17* and *NANOG* boundary knockout constructs were deposited on addgene.org under ID #172225 and ID #172224 respectively.

**Cell culture and CRISPR/Cas9 targeting.** mTeSR1 (Stemcell Technologies) maintained ZIP13K2 (ref. [31]) human induced pluripotent stem cells were treated with Accutase (Sigma–Aldrich, A6964), supplemented by 10 μM Y-27632 (Tocris, 1254) for 15 min at 37 °C, 5% $CO_2$ to obtain single cells. To quench and wash the cells, equal volumes of mTeSR1were added and cells spun down for 5 min at 300 × g, 21 °C. Cells were further seeded in mTeSR1 containing 10 μM Y-27632 at a density of $3 \times 10^5$ /cm$^2$ on Matrigel (Corning) precoated six-well plates (Corning) and cultured 16–24 h at 37 °C, 5% $CO_2$ before transfection. Transfection was carried out with up to 5 μg of modified P2X458 (including both respective sgRNAs) using Lipofectamine 3000 (Thermo Fischer Scientific) according to the manufacturers protocol. GFP$^+$ cells were FACS-sorted 16–24 h post-transection on the FACS Aria II (Beckton Dickinson) and seeded in low density ($5–10 \times 10^5$/55 cm$^2$) using mTeSR1 supplemented with 10 μM Y-27632 (Tocris, 1254) to derive isogenic clones. Single-cell-derived colonies were picked, and half kept for maintenance respectively used for genotyping with the Phire Animal Tissue Direct PCR Kit (Thermo Fischer Scientific) accordingly. Genotypes were verified by cloning QIAquick Gel Extraction Kit (Quiagen) purified PCR products (Supplementary Table 2) into the pJET1.2 backbone (Thermo Fischer Scientific) and sanger sequencing of PCR single-products was performed with at least 10× positively transformed 10-beta *E. coli* (NEB, C3019H) colonies.

**Endoderm differentiation, DKK4 inhibition and DKK1 treatment.** ZIP13K2 cultures were treated with Accutase supplemented by 10 μM Y-27632to obtain single cells. To quench and wash the cells, equal volumes of mTeSR1were added and cells spun down for 5 min. at 300 × g, 21 °C. After resuspension in mTeSR1 supplemented by 10 μM Y-27632, cells were counted and seeded according to the manufacturer's instructions on Matrigel (Corning) precoated culture plates/dishes. Media change using the STEMdiff Trilineage Endoderm Differentiation media was performed on a daily base after washing the cultures with equal volumes of DPBS (Thermo Fischer Scientific, 14190250) according to the manufacturer's instructions. In the case of DKK4 inhibition, differentiation media was supplemented with 50 μM DKK4 inhibitor 9-Carboxy-3-(dimethyliminio)-6,7-dihydroxy-10-methyl-3H-phenoxazin-10-ium iodide (Merck, 317701). In the case of DKK1 treatment, differentiation media was supplemented with 150 ng/ml recombinant human DKK1 (R&D Systems, 5439-DK-010/CF).

**DKK4 enzyme-linked immunosorbent assay (ELISA).** Cell culture media supernatants from undifferentiated or differentiated cells across several timepoints (see *Endoderm differentiation*) of different cell lines were collected, spun at 300 × g, 5 min at 4 °C. Cell free supernatants were again collected and snap frozen at −80 °C in dry ice. Prior to ELISA, supernatants were thawed on ice and prediluted 1:200 in reagent diluent (R&D Systems, DY995) of the Human Dkk4 DuoSet ELISA KIT (R&D Systems, DY1269). DKK4 ELISA has been carried out according to the manufacturer's instructions. Cell culture media supernatants from undifferentiated or differentiated cells across several timepoints (see. *Endoderm differentiation*) of different cell lines were collected, spun at 300 × g, 5 min at 4 °C. Cell free supernatants were again collected and snap frozen at −80 °C in dry ice. Prior to ELISA, supernatants were thawed on ice and prediluted 1:200 in reagent diluent

 

(R&D Systems, DY995) of the Human Dkk4 DuoSet ELISA KIT (R&D Systems, DY1269). DKK4 ELISA has been carried out according to the manufacturer's instructions. HRP raw values were measured on the GloMax-Multi Detection System (Promega).

**FACS and Immunofluorescence staining**. Undifferentiated or differentiated ZIP13K2 cultures were treated with Accutase (Sigma–Aldrich, A6964) to obtain single cells. To quench and wash the cells, suspensions were supplemented with FACS buffer containing final 5 mM EDTA (ThermoFischer Scientific, 15575020), 10% fetal bovine serum (FBS) (ThermoFischer Scientific, 26140079) in 1× DPBS (Thermo Fischer Scientific, 14190250). Further, cells were washed and surface stained in FACS buffer for 30 min at 4 °C using antibody dilutions according to the manufacturer's instructions with slight modifications (Supplementary Table 3). Cells were again washed as described above, fixed and intracellularly stained utilizing the True-Nuclear™ Transcription Factor Buffer Set (Biolegend, 424401) according to manufacturer's instructions. Following subsequent wash steps in permeabilization buffer, we performed flow cytometry data acquisition on the Celesta (Beckton Dickinson, IC-Nr.: 68186, Serial-Nr.: R66034500035). Raw data were analyzed by the use of FlowJo (Beckton Dickinson) v10.7.2.

Undifferentiated or differentiated cell cultures for immunofluorescent (IF) stainings were directly fixed on the culture plates, using 4% PFA solution in DPBS for 15 min at 21 °C. Followed by multiple wash steps with DPBS, cultures were permeabilized in PBT-buffer containing 1% BSA (Sigma–Aldrich, A2153), 10% FBS (ThermoFischer Scientific, 26140079) and 0,3% Triton-X-100 (Sigma–Aldrich, T8787) in DPBS for 30 min at 21 °C. Blocking was further performed in PB buffer (PBT without Triton-X-100) for 30 min at 21 °C. Subsequently, cultures were washed in DPBS and incubated with primary or secondary antibody solutions for at least 2 h at 21 °C (Supplementary Table 4). DNA staining was performed using 0.25 µg/ml DAPI solution (ThermoFischer Scientific, D1306) for 15 min at 21 °C. Microscopy was performed using the Z1 Observer (Zeiss) and fluorescent raw signals were adjusted according to the respective controls using ZEN 2 blue (Zeiss) V2.3. Cell quantification and MFI measurements of the respective channel were performed using Fiji (65, 255 threshold; watershed function; 0.05-0.50 particle size).

**Generation of a polyclonal SOX17-TagBFP cell line and rescue of endogenous SOX17 protein**. PB-CAG-DD-3xFLAG-hSOX17-GS-TagBFP-BGHpA rescue construct was generated by Gibson Assembly® (NEB, E2621L) of BstBI /BamHI double-digested PB-CAG-BGHpA (Addgene Plasmid #92161) and EcoRI digested synthetically generated pUC19 DD-3xFLAG-SOX17-GS-TagBFP (Genewiz). PB-CAG-BGHpA was a gift from Xiaohua Shen (Addgene plasmid # 92161; http://n2t.net/addgene:92161; RRID:Addgene_92161)[70]. PB-CAG-DD-3xFLAG-hSOX17-GS-TagBFP-BGHpA rescue construct was deposited on addgene.org under ID #172226. Both, PB-CAG-DD-3xFLAG-hSOX17-GS-TagBFP-BGHpA and Super PiggyBac transposase expression vector (SBI, PB210PA-1) were co-transfected into SOX17Δ5'CTCF#8.2 mTeSR1 (Stemcell Technologies) maintained human induced pluripotent stem cells harboring the SOX17 boundary 2 deletion. Transfection was conducted using equimolar plasmid ratios in combination with Lipofectamine Stem Transfection Reagent (Thermo Fischer Scientific, STEM00003) according to the manufacturer's instructions. Transfected or untransfected cells were treated with mTeSR1 (Stemcell Technologies) containing 250 µg/ml m Hygromycin B (Carl Roth, 1287.1) for 2 weeks. TagBFP-negative surviving cells were FACS-sorted on the FACS Aria Fusion (Beckton Dickinson) and seeded in low density (5–10 × 10⁵/55 cm²) using mTeSR1 supplemented with 10 µM Y-27632 (Tocris, 1254) to derive a polygenic/polyclonal SOX17 rescue cell line. To stabilize ectopic SOX17-TagBFP protein, undifferentiated iPSC or day 2 dEN onwards differentiating cells were treated with 1 µM final Shield-1 (Takara, 632189) back to back with untreated controls before sample collection for downstream analysis.

**Western Blot**. Undifferentiated or differentiated ZIP13K2 cultures were treated with Accutase for 15 min, 37 °C, 5% CO2 to obtain a single suspension. Single-cell suspensions were washed once with ice cold DPBS and spun down at 300 × g, 5 min at 4 °C. Supernatants were removed and cell lysates generated using treatment for 30 min on ice with RIPA buffer (Thermo Fisher Scientific) supplemented with 1 × HALT protease inhibitor (Thermo Fisher Scientific, 87786). Lysates were spun down at 12,000 × g, 10 min at 4 °C and supernatants quantified for protein content using the Pierce BCA Protein Assay Kit (Thermo Fisher Scientific, 23227) according to the manufacturer's instructions. For western blot, 10 µg total protein extract per sample were boiled in final 1 × Laemmli Buffer (BioRad, 1610747) containing 10% 2-Mercaptoethanol (Sigma–Aldrich) for 10 min at 95 °C, followed by cooling on ice for 5 min. Samples were then loaded on a NuPAGE 4–12%, Bis-Tris, 1,0 mm, Mini Protein Gel (Thermo Fisher Scientific, NP0322BOX) and ran at 200 V for 30 min in 1 × NuPAGE MOPS SDS Running Buffer (Thermo Fisher Scientific, NP0001) containing 1:400 NuPAGE Antioxidant (Thermo Fisher Scientific, NP0005). Protein transfer has been performed utilizing the iBlot 2 Starter Kit, PVDF (Thermo Fisher Scientific, IB21002S) following the manufacturer's instructions for the P0 program. PVDF membranes containing transferred proteins were incubated in blocking buffer (1 × TBS-T (Thermo Fisher Scientific), 5% Blotting-Grade Blocker (BioRad, 1706404)) for 1 h at RT. Incubation with primary antibody dilution (see below) was performed in blocking buffer at 4 °C overnight.

The following day, membranes were washed three times 10 min at RT with 1 × TBS-T and incubated for 2 h at RT in secondary antibody dilution in blocking buffer (Supplementary Table 5). Next, membranes were washed three times for 10 min at RT with 1 × TBS-T and developed using the SuperSignal West Dura Extended Duration Substrate (Thermo Fisher Scientific, 34075) according to the manufacturer's instructions on the BioRad ChemiDoc XRS+ imaging system.

**SureSelect design**. The library of SureSelect enrichment probes were designed over the genomic interval (hg19, chr8:54735936-55657612) using the SureDesign online tool of Agilent. 3299 total probes cover the SOX17 locus and were designed to specifically enrich for regions in proximity of NlaIII sites. The probes covered 35.25% of the interval.

**Capture Hi-C (cHi-C) sequencing and data analysis**. cHi-C libraries were prepared from wild-type or homozygous SOX17Δ5'CTCF iPSC or dEN cells. Undifferentiated or day 5 differentiated ZIP13K2[31] cells were grown to a final count of 4–5 million, treated with Accutase (Sigma–Aldrich, A6964), resuspended and washed in DPBS. Cell lysis, NlaIII (NEB R0125) digestion, ligation, and decrosslinking were performed according to the Franke et al. protocol[15]. Adaptors were added to DNA and amplified according to Agilent instructions for Illumina sequencing. The library was hybridized to the custom-designed sure-select beads and indexed for sequencing of 200 × 10⁶ fragments per sample (100 bp paired-end) following the Agilent instructions. Capture Hi-C experiments were performed as biological duplicates.

Raw sequence reads of capture Hi-C (cHi-C) were mapped to the hg19 version of the human genome using BWA (v0.7.17) with parameters (mem -A 1 -B 4 -E 50 -L 0). Mapped reads were further processed by HiCExplorer (v3.5.1) to remove duplicated reads and reads from dangling ends, self-circle, self-ligation, and same fragments. The replicates of the same samples were compared, and confirmed to have high consistency (Pearson correlation coefficient: 0.83–0.99), then were merged to construct contact matrices of 2 kb resolution. Normalization was performed to ensure that all samples have the same number of total contacts, followed by KR normalization. The relative contact difference between two cHi-C maps was calculated by subtracting one from the other after scaling one sample to the other by using the total number of contacts in each sample.

**RNA-sequencing and data analysis**. Triplicates of either undifferentiated or differentiated ZIP13K2[31] cultures were treated with Accutase (Sigma–Aldrich, A6964) and differentiated cultures were further quenched with FACS buffer containing 5 mM EDTA (ThermoFischer Scientific, 15575020) 10% FBS (Thermo-Fischer Scientific, 26140079) in DPBS (Thermo Fischer Scientific, 14190250) to obtain single cells. In order to enrich for CXCR4⁻ or CXCR4⁺ cell fractions of differentiated cultures, cells were stained for anti-Human CRCX4 (CD184) PE (as described under 21. FACS) and compared to Isotype and unstained control sorted for either CXCR4⁻ or CXCR4⁺ subpopulations on the Aria II (Beckton Dickinson). RNA isolation including on-column DNase digest of enriched cell populations was performed using the RNeasy Mini Kit (Qiagen, 74104) according to the manufacturer's instructions. The KAPA Stranded mRNA-Seq Kit (Kapa Biosystems, #KK8401) was utilized for RNA library preparation, using 500 ng total RNA and performing poly-(A) enrichment followed by first-strand cDNA-synthesis (11 cycles). Subsequently, RNA-sequencing libraries were prepared by the use of dual index primers according to the manufacturer's instructions. Illumina adapter ligated sequencing libraries were sequenced for 50 million 75 bp long read pairs per sample on the HiSeq4000 (Illumina). RNA-seq data were preprocessed using cutadapt[71] to remove adapter sequences and trim low-quality bases. Reads were aligned against hg19 using STAR[72] (v 2.6.1d, parameter:–outSAMtype BAM SortedByCoordinate–outSAMattributes Standard–outSAMstrandField intronMotif–outSAMunmapped Within–quantMode GeneCounts). Subsequently, Stringtie[73] (v 1.3.5) was used for transcript assembly, e.g., calculation of strand-specific TPMs. Differential expression analysis was done independently per group comparison using the R package DESeq2[74] utilizing the raw expression counts from STAR's reads per gene output and filtered for an adjusted $p$-value < 0.05 and a log2 fold-change > 1. The PCA was calculated on the log2+1 normalized TPMs of the 100 most variable genes using the R function prcomp (parameters "center = TRUE, scale = TRUE"). Box- and scatter plots show the unmodified TPMs. The heatmaps shows Z-score normalized TPMs to adjust for differences in absolute expression levels and was plotted using the R package pheatmap.

**qRT-PCR gene expression**. TaqMan-based qRT-PCR reactions were set up in triplicate using the 2× TaqMan Fast Advanced Master Mix (Thermo, 4444557) according to manufacturer's instructions. Reactions were run on a StepOnePlus (Thermo) PCR machine with 40 cycles of 1 s at 95 °C and 20 s at 60 °C. ollowing TaqMan probes (Thermo) were used: SOX17 Hs00751752_s1; NANOG Hs02387400_g1; T/BRACHYURY Hs00610080_m1; FOXA2 Hs00232764_m1; GATA4 Hs00171403_m1; DKK1 Hs00183740_m1; DKK2 Hs00205294_m1; DKK4 Hs00205290_m1; 18 s Hs03003631_g1. NANOGNB Hs04225119_g1; GDF3 Hs00220998_m1; APOBEC1 Hs00242340_m1; DPPA3 Hs01931905_g1; CLEC4C Hs01092460_m1; ATP6V1H Hs00977530_m1; RGS20 Hs00991569_m1; TCEA1 Hs04403253_g1; LYPLA1 Hs00911024_m1; MRPL15 Hs00204356_m1; RP1 Hs00196698_m1.

**ChIP qRT-PCR**. For CTCF-ChIP qRT-PCR, undifferentiated ZIP13K2[31] cells were grown to a final count of 10 million, treated with Accutase (Sigma–Aldrich, A6964), resuspended and washed in DPBS. Subsequently, cells were crosslinked in 1% formaldehyde solution for 5 min at room temperature. Following quenching with 0,125 M glycine final and two DPBS washes, we isolated nuclei using 1 ml cell lysis buffer (20 mM Tris-HCl ph8.0, 85 mM KCl, 0.5% NP40) for 10 min on ice. Then nuclei were spun down for 3 min at $2500 \times g$ and supernatant was removed. The pellet was resuspended in 1 ml of nuclear lysis buffer (10 mM Tris-HCl, pH 7.5, 1% NP40, 0.5% sodiumdeoxycholate, 0.1% SDS) then incubated for 10 min on ice. Sonication was carried out on a Covaris E220 Evolution sonicator (PIP = 140.0, Duty Factor = 5.0, Cycles/Burst = 200, 10 min). After sonication, chromatin was spun down at $15,000 \times g$ for 10 min to pellet insoluble material. Volume was increased to 1,5 mL with Chip Dilution Buffer (0.01%SDS, 1.1% Triton-X-100, 1.2 mM EDTA, 16.7 mM Tris-HCl pH 8.1, 167 mM NaCl), and 20 μl of CTCF antibody (CST, D31H2-XP) was added. Immunoprecipitation mixture was allowed to rotate overnight at 4 °C. The next day, 40 μl of Protein A Dynabeads (Thermo, 10001D) were added to the IP mixture and allowed to rotate for 4 h at 4 °C. This was followed by two washes of each: low salt wash buffer (0.1% SDS, 1% Triton-X-100, 2 mM EDTA, 20 mM Tris-HCl pH 8.1,150 mM NaCl); high salt wash buffer (0.1% SDS, 1% Triton-X-100, 2 mM EDTA, 20 mM Tris, pH 8.1, 500 mM NaCl); LiCl wash buffer (0.25 M LCl, 1% NP40, 1% deoxycholate, 1 mM EDTA, 10 mM Tris-HCl pH 8.1); and TE buffer pH 8.0 (10 mM Tris-HCl, pH 8.0, 1mMEDTA pH 8.0). DNA was eluted twice using 50 μl of elution buffer (0.5–1% SDS and 0.1 M NaHCO3) at 65 °C for 15 min. 16 μl of reverse crosslinking salt mixture (250 mM Tris-HCl, pH 6.5, 62.5 mM EDTA pH 8.0, 1.25 M NaCl, 5 mg/ml Proteinase K) was added and samples were allowed to incubate at 65 °C overnight. DNA was purified using AMPure XP beads (Beck-man-Coulter) and treated with DNase-free RNase (Roche) for 30 min at 37 °C.

qRT-PCR reactions were set up in triplicate with the 2× PowerUp SYBR Green Master Mix (Thermo, A25742). Reactions were run on a StepOnePlus (Thermo) PCR machine with 40 cycles of 15 s at 95 °C and 60 s at 60 °C (Supplementary Table 6).

## Data availability

The data that support this study are available from the corresponding authors upon reasonable request. All Hi-C, RNA-seq, and capture Hi-C data generated in this study have been deposited in the NCBI Gene Expression Omnibus (GEO) database under accession number GSE127196. The Hi-C data used in this study are available in the GEO database under accession number GSE52457 and GSE63525. Hi-C data of human embryos were obtained from the Genome Sequence Archive with the accession number CRA000852. CTCF-ChIP-seq data used in this study are available at Cistrome (http://cistrome.org) and in the GEO database under accession number GSM518375, GSM325897, GSM614637, GSM614636, GSM614631, GSM614630, GSM325899, GSM614615, GSM614614, GSM651543, GSM651542, GSM651541, GSM586888, GSM586887, GSM534492, GSM534485, GSM534478, GSM534471, GSM325895, GSM489290, GSM489291, GSM489292, GSM489293, GSM489294, GSM489295, GSM489296, GSM489297, GSM489298, GSM489299, GSM489300, GSM489301, GSM489302, GSM782156, GSM782158, GSM631475, GSM631476, GSM631477, GSM631478, GSM631479, GSM624077, GSM624078, GSM624079, GSM624080, GSM624081, GSM748538, GSM748539, GSM941710, GSM1056576, GSM1056577, GSM1070125, GSM646475, GSM646412, GSM646455, GSM646413, GSM646454, GSM646474, GSM646432, GSM646433, GSM1138985, GSM822276, GSM822271, GSM822277, GSM822297, GSM822299, GSM822294, GSM822278, GSM1007997, GSM1007998, GSM646373, GSM646315, GSM646334, GSM646353, GSM646372, GSM646354, GSM646314, GSM646335, GSM646392, GSM646393, GSM808772, GSM808759, GSM808771, GSM808752, GSM808764, GSM808765, GSM808753, GSM808760, GSM1208603, GSM947527, GSM947528, GSM849300, GSM849301, GSM849304, GSM849302, GSM849305, GSM489305, GSM970828, GSM1055825, GSM1224649, GSM1224650, GSM1224651, GSM1224652, GSM1224653, GSM1224654, GSM1224655, GSM1224656, GSM1224657, GSM1224658, GSM1224659, GSM1224660, GSM1233869, GSM1233870, GSM1233887, GSM1233888, GSM1233914, GSM1233915, GSM1233916, GSM1233933, GSM1233934, GSM1233955, GSM1233956, GSM1233977, GSM1233978, GSM1233979, GSM1233993, GSM1233994, GSM1234010, GSM1234011, GSM1234027, GSM1234028, GSM1234044, GSM1234045, GSM1234061, GSM1234062, GSM1234078, GSM1234079, GSM1234099, GSM1234100, GSM1234121, GSM1234122, GSM1234144, GSM1234145, GSM1234146, GSM1234162, GSM1234163, GSM1234164, GSM1234180, GSM1234181, GSM1234182, GSM1234198, GSM1234199, GSM1234200, GSM1234216, GSM1234217, GSM1234218, GSM1234219, GSM1239390, GSM1239588, GSM1240813, GSM1240827, GSM1335528, GSM1003582, GSM1003581, GSM1003474, GSM1003464, GSM733752, GSM733672, GSM733785, GSM733645, GSM733724, GSM733762, GSM733783, GSM733716, GSM733719, GSM1003508, GSM733765, GSM733744, GSM733636, GSM733695, GSM733784, GSM1022640, GSM1022639, GSM1003606, GSM822289, GSM749678, GSM749695, GSM749769, GSM1022635, GSM749750, GSM749680, GSM749728, GSM749723, GSM749714, GSM749759, GSM749736, GSM749666, GSM1022653, GSM1022650, GSM749752, GSM749677, GSM749698, GSM749748, GSM749708, GSM749709, GSM749705, GSM1006891, GSM1006870, GSM749711, GSM1022664, GSM749762, GSM749676, GSM749725, GSM1022636, GSM749740, GSM1022633, GSM749692, GSM749694, GSM1022629, GSM749686, GSM749730, GSM749741, GSM749757, GSM749670, GSM749764, GSM749706, GSM749704, GSM935611, GSM822312, GSM1006885, GSM1006869, GSM1006873, GSM1022668, GSM749696, GSM1022662, GSM1022661, GSM749710, GSM749743, GSM749732, GSM1022657, GSM1022677, GSM749745, GSM749735, GSM1022652, GSM1022651, GSM749726, GSM749712, GSM749668, GSM749687, GSM749729, GSM749739, GSM822285, GSM749715, GSM749683, GSM822287, GSM1022644, GSM1022671, GSM1022669, GSM749688, GSM1022631, GSM749753, GSM749665, GSM749675, GSM749751, GSM749681, GSM749699, GSM749717, GSM749737, GSM749727, GSM749673, GSM1022665, GSM749749, GSM749674, GSM822279, GSM1022630, GSM1022628, GSM935404, GSM822311, GSM935407, GSM749690, GSM749733, GSM1006886, GSM1006887, GSM1006874, GSM1006882, GSM822305, GSM1006875, GSM1022663, GSM1022658, GSM1006878, GSM822308, GSM822309, GSM1006893, GSM1022643, GSM1022675, GSM1022676, GSM749747, GSM749707, GSM1022626, GSM1006881, GSM1022666, GSM1022667, GSM749779, GSM749684, GSM1003633, GSM749693, GSM749667, GSM1006883, GSM749768, GSM749679, GSM1022634, GSM1022637, GSM803453, GSM803456, GSM1010774, GSM803419, GSM1010903, GSM803486, GSM1010820, GSM1010734, GSM803333, GSM803348, GSM1122667, GSM1224672, GSM1224673, GSM1224674, GSM1224675, GSM1273199, GSM1294054, GSM1294055, GSM1354438, GSM1354439, GSM1383877, GSM1463918, GSM1267206, GSM1267207, GSM1267208, GSM1267209, GSM1267210, GSM1505620, GSM1505623, GSM1505624, GSM1505625, GSM1505626, GSM1689152, GSM1782700, GSM1782701, GSM1817654, GSM1817658, GSM1817662, GSM1817665, GSM1817666, GSM1817667, GSM1910997, GSM1910998, GSM1684571, GSM1684572, GSM1684573, GSM1684574, GSM1705253, GSM1705254, GSM1705262, and GSM1705263. RNA-seq and H3K27ac ChIP-seq data used in this study are available at Epigenomics Roadmap Project (http://www.roadmapepigenomics.org). Enhancers used in the study are available at Fantom5 database (https://fantom.gsc.riken.jp/5) and Epigenomics Roadmap Project (http://www.roadmapepigenomics.org). Other data supporting the findings of this study are included in the paper and the supplementary file. Source data are provided with this paper.

## Code availability

The computational code used in the manuscript is available at https://bitbucket.org/mthjwu/loop_cluster.

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

## Acknowledgements

We would like to thank the Michor and Meissner labs for helpful discussions as well as Bernd Timmermann and the MPIMG sequencing core facility for the ongoing support. We gratefully acknowledge support of the Dana-Farber Cancer Institute Physical Sciences-Oncology Center, NIH U54CA193461 (to F.M.), NIH 1P50HG006193, P01GM099117, 1DP3K111898 (to A.M.), and the Max Planck Society (to A.M.). We acknowledge support of the High-performance Computing Platform of Peking University.

## Author contributions

H.J.W. conceived, planned, designed, and performed the data analyses. A.L. conceived, planned, designed, and performed the experiments. E.S. and A.B. helped in performing experiments. H.K. performed the bioinformatics analysis of RNA seq. A.M. and F.M. conceptualized, designed, and supervised the study. H.J.W., A.L., A.M., and F.M. wrote the manuscript. All authors contributed to the final paper.

## Competing interests

A.M. and F.M. are co-founders of an oncology company. The remaining authors declare no competing interests.
