## [Peer Review File · Nature Communications]

REVIEWERS' COMMENTS

Reviewer #1 (Remarks to the Author):

The authors have addressed my comments. They have submitted a substantially improved manuscript. I recommend this paper for publication.

Reviewer #2 (Remarks to the Author):

This is a revised version of a paper that was initially submitted two years ago. There are significant changes to the manuscript and the quality of the data is improved. The novelty aspect is still lacking, especially now with additional studies having appeared that perturbed CTCF sites or domain boundaries. Perhaps genes encoding “developmental regulators” tend to be slightly more insulated than other genes, but I am not really sure whether their genome organization is really categorically different.

Specific comments:

1. line 71: “most studies focused on developmental processes in somatic cells; however, embryonic development is arguably more vulnerable...” This is an odd comment since most embryonic tissues are somatic, and the studies mentioned in the introduction affect embryonic development such as limb formation etc.
2. It is difficult to make broad inferences when perturbing only two loci (Sox17 and NANOG). When generalizing based on computational analysis of contact maps, it is important to remember that just because a looped domain is detectable does not mean the loop matters. In this paper (Splinter, G&D 2006), one of the first of its kind, deleting CTCF sites or depleting CTCF altered loops but not gene expression. Loop domains can form in the absence of internal contacts and vice versa (e.g. Brown, Nat Comm 2018).
3. There is a tendency to play up low p values to build the story but not considering modest biological effect sizes. The differences in intensity, preservation, and a-amanitin response between TIG boundary contacts and multi-gene boundary contacts is not very strong (Fig.2C,E). The difference between in CTCF binding strength between TIG-, non-TIG, or zero-gene boundaries and conservation scores is modest at best (Fig.3d,S3c). Similarly, is a 1.8-fold enrichment in a very broad GO term (development) biologically significant such that TIGs represent a new entity?
4. I am not sure this is an unexpected insight: “ Further analyses demonstrated that enhancer activity is more correlated with its neighbor gene expression within constitutive domains than outside of them” or the conclusion that “These results underscore the importance of insulation”.